# Mechanochemical feedback between confinement and actin crosslinking drives the shape dynamics of liquid-like droplets

Daniel Mansour [1], Dominique Jordan [2], Caleb Walker[2], Aravind Chandrasekaran[1], Christopher T. Lee [3], Kristin Graham [2], Jeanne C. Stachowiak [2,4] ✉ & Padmini Rangamani [1,5] ✉

Several actin-binding proteins form phase-separated condensates that promote actin filament assembly and bundling. However, the mechanism by which crosslinker multivalency, actin growth, and condensate mechanics regulate actin organization and droplet shape is not well understood. Here, using a combination of agent-based simulations and experiments, we show that a dynamically deformable droplet interface enables the emergence of tightly-bundled actin rings and weakly-bundled actin discs. We find that crosslinked bundle thickness and droplet diameter follow a power law, consistent with measurements in condensates formed by vasodilator-stimulated phosphoprotein. In addition, the dynamics of droplet deformation exhibit a dynamic snapping behavior that depends on droplet surface tension and crosslinker binding kinetics. We assess the generalizability of these predictions in condensates formed by lamellipodin and RGG. Together, these results indicate that mechanochemical feedback between droplet interfacial mechanics and crosslinker multivalency tunes actin organization and controls the dynamics of droplet deformation driven by actin networks.

Biomolecular condensates form membraneless organelles that function as dynamic compartments for organizing various cellular functions[1–3]. Recent discoveries have shown that several actin-binding proteins (ABPs) can form condensates, through liquid-liquid phase separation, both in vitro and in vivo[3–12]. Over 160 ABPs have been identified, and each of them plays a distinct role in transforming actin monomers and unorganized filaments into specialized structures that drive complex biological processes[13,14]. Despite their diversity, ABPs collectively regulate actin filaments through behaviors that can be broadly classified as tuning filament assembly and disassembly rates, branching, severing, annealing, and bundling, which leads to the formation of a variety of actin network architectures. These structures provide the mechanical framework for filopodia, lamellipodia, stress fibers, focal adhesions, contractile rings, and the actin cortex[4,13,15,16]. The reorganization of the actin cytoskeleton underlies a wide range of cellular processes that require cells to generate and transmit force, change shape, and respond to mechanical and chemical cues in their environment, including motility, signaling, and intracellular transport[10,17–21]. Dysregulation of these processes is implicated in many pathologies, notably in cancer and metastasis[20]. Importantly, ABP condensates have been observed in cells and are thought to provide a favorable environment for the recruitment of actin monomers, actin filaments, and other cytoskeletal proteins, and to promote nucleation of actin filaments[12,16,22–27]. However, a comprehensive understanding of the role of ABP condensates in actin network organization and remodeling is an active area of investigation.

[1]Department of Mechanical and Aerospace Engineering, University of California San Diego, La Jolla, CA, USA. [2]Biomedical Engineering, The University of Texas at Austin, Austin, TX, USA. [3]Department of Molecular Biology, University of California San Diego, La Jolla, CA, USA. [4]Chemical Engineering, The University of Texas at Austin, Austin, TX, USA. [5]Department of Pharmacology, University of California San Diego School of Medicine, La Jolla, CA, USA. ✉e-mail: jcstach@austin.utexas.edu; prangamani@ucsd.edu

Due to the high complexity of in vivo condensate assemblies, reconstituted systems have been employed to study interactions between actin and ABP condensates in a controlled environment to understand the underlying biophysical principles of actin organization and confinement[28]. Similar actin ring formation has been observed in experimental studies using giant unilamellar vesicles (GUVs)[29,30] and vesicles[31–33]. Beyond the biochemical function of the constituent ABPs, the material and physical properties of condensates also regulate their function[24,34,35]. The same weak interactions that drive condensate formation also allow for the development of cohesive forces that endow the liquid droplet with viscoelastic properties such as surface tension and viscosity[24,35–37]. For actin filaments already within condensates, the surface tension drives confinement, which supports the accumulation of bending energy[5,6,38]. Condensate deformation occurs once the filament bending energy exceeds the surface energy[6,39]. Bending energy may accumulate from a combination of actin polymerization and increases in actin bundle rigidity mediated by ABP processes such as actin crosslinking. The resulting interplay between surface energy and bending energy plays a key role in regulating the resulting structures.

By combining in vitro and in silico approaches, we have been able to create a complementary framework where experiments provide physical constraints and simulations systematically explore the underlying mechanisms and generate new experimentally testable hypotheses. We have previously established how the spatial confinement of ABPs through phase separation affects actin filament assembly and bundling[5,6,39,40]. For example, we recently showed that vasodilator-stimulated phosphoprotein (VASP) forms phase-separated droplets wherein monomeric actin can assemble into filamentous actin networks[6]. Depending on the ratio of actin to VASP, the resulting actin networks can form shells, rings, or ellipsoidal discs[6,39]. Actin ring structures are tightly packed and aligned into a nearly 2D distribution, while actin disc structures are loosely aligned into a thick ellipsoidal structure. The formation of crosslinked actin networks (rings) was a precursor to droplet deformation, where spherical VASP droplets transitioned to ellipsoidal droplets and finally rods[6,39]. More recently, we also showed experimentally that actin bundling did not require native tetrameric strong bundling proteins such as VASP but was also possible due to proteins such as lamellipodin (Lpd), which bind actin but with a lower binding valency and without inherent polymerase activity[16,41–43]. We also showed that when Eps15, a condensate-forming protein without any known interaction with actin, is fused with Lifeact, a general F-actin binding motif, the resulting Eps15-Lifeact composite promotes actin filament assembly and bundling[5]. Using an agent-based modeling approach, we investigated how interactions between VASP and actin in rigid, non-deformable droplets can lead to the formation of rings versus shells[39]. We showed that rings of actin resulted from kinetic trapping, where, for a fixed filament growth rate, the binding and unbinding rate of VASP to actin allowed sufficient VASP residence time for binding and zippering additional filaments[6]. We also showed that systems with dynamic dimers and multimers, corresponding to Lpd and Eps15-Lifeact composites, respectively, formed rings versus shells resulting from kinetic trapping[5]. In the case of GUV-encapsulated actin, membrane-actin tethering is critical for ring formation[31]. These experimental observations suggest that general principles govern the assembly of actin shells, rings, and discs in protein droplets.

However, two questions remain: What is the role of mechanical feedback from deformable boundary confinement in tuning actin organization within the droplet? How do the biochemical properties of the crosslinker influence droplet deformation? To address these questions, we built a minimal model that captures the key elements of actin organization in phase-separated droplets using a combination of computational and experimental approaches. We modeled the ABP droplet using dynamically deformable ellipsoidal boundaries and represented both the ABP-ABP and ABP-actin kinetic interactions explicitly (Fig. 1). Our simulations revealed that the condensate environment tunes the nature of actin organization through kinetic trapping, thereby controlling droplet shape. We predicted that the droplet diameter scales with the actin bundle thickness required to deform a spherical droplet according to a power law. These predictions were validated by experiments. We then showed in simulations and experiments that capping protein tunes the filament length distribution and can alter the onset and extent of VASP droplet deformation. Finally, we characterized our observations throughout this study, which showed that the temporal behavior of droplet deformation has a dynamic snapping behavior depending on the droplet mechanical properties and the crosslinker kinetic properties. Thus, our study produces a comprehensive, general framework to understand the relationship between droplet deformation and the mechanochemical environment of the droplet.

## Results

We constructed an agent-based model to investigate the interplay between the physical and chemical properties of the ABP droplet and the actin network. For these simulations, we chose to use the Cytosim framework, which simulates the chemical dynamics and mechanical properties of cytoskeletal filament networks[44]. This modeling framework allows us to explicitly simulate three distinct components: filaments, the droplet space, and crosslinkers.

Actin filaments are represented as inextensible fibers that elongate at a constant rate due to polymerization (Fig. 1A). We assume that the plus end elongates while the minus end is stable. These assumptions are consistent with the activity of VASP as a processive polymerase that mediates up to a three-fold increase in elongation rate at bound barbed ends compared to free barbed ends[45], and based on previous models of actin filament elongation with a net elongation at the plus ends[39]. In this study, we introduce a method for implicitly modeling the effect of capping protein by stochastically stopping (capped state) and restarting (uncapped state) elongation via a capping reaction governed by the rate parameters $k_{cap}$ and $k_{uncap}$ (Fig. 1B). Finally, actin filaments are modeled as a series of linear segments of length 0.1 μm that are connected by hinge points to allow for bending (Fig. 1C). The bending stiffness of the fiber is captured by the flexural rigidity parameter $k_{bend}$[46].

The ABPs are represented as spherical solids with a radius of 30 nm, with one of two different kinds of binding sites that either bind actin to promote bundling or bind to other ABP molecules to promote multimerization. Each binding site has a specified binding rate constant $k_{bind}$ and binding distance [30 nm] that defines the radius within which targets are considered in range for binding to occur[39]. The unbinding rate, $k_{unbind}$, specifies the rate constant used, while the unbinding force $F_0$ represents the characteristic force used in the force-dependent kinetics (Bell's law and Kramers' theory) representation of slip bond unbinding kinetics[47–52]. We investigated the role of two distinct types of actin crosslinkers: tetrameric actin crosslinkers representing VASP[39,40] and dynamically multimerizing actin crosslinkers representing the transient multivalent interactions between monomeric ABPs in the condensate environment[5]. VASP is modeled with four actin-binding sites that enable it to undergo a series of reversible reactions between five states that differ depending on the number of bound actin filaments (Fig. 1D)[39]. Additionally, we included modifications to the Cytosim codebase to model dynamically multimerizing monomers based on a specified multimer formation rate $k_{form}$ and splitting rate $k_{split}$, as introduced in Walker et al.[5]. These dynamic crosslinkers differ from static crosslinkers in that each crosslinker molecule is a monomer modeled with a single actin-binding site and two monomer-monomer binding sites to allow each monomer to bind with up to two other monomers to dynamically form multimers (such as dimers, trimers, tetramers, etc.). Due to the large number of possible states, we categorize each monomer unit by the number of other monomers it is bound to [0, 1, 2] and by whether or

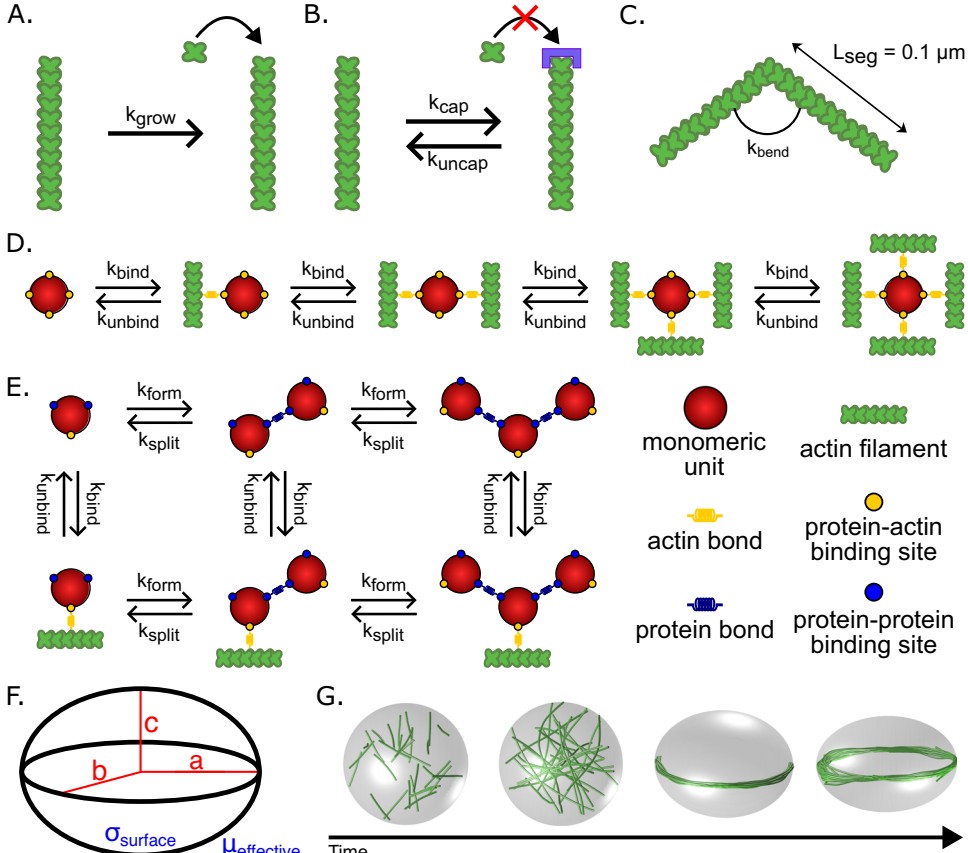

**Fig. 1 | Simulation framework for capturing actin network properties in droplets of actin-binding proteins (ABPs). A** Actin filaments are modeled with a deterministic growth rate $k_{grow}$ that was chosen to allow the filaments to grow to the length of the circumference of the starting spherical droplet space by the end of the simulation (600 s). The growth rate is set at $k_{grow} = 0.0103$ μm/s unless otherwise stated. **B** Capping protein is implicitly modeled using rates of capping and uncapping that are specified at the plus (+) ends of the actin filaments. Capping of filament ends halts their growth. **C** Actin filaments are modeled as inextensible segments of length $L_{seg}$ that can bend along hinge points based on flexural rigidity $k_{bend}$. **D** Schematic of the different combinations of possible binding and unbinding reactions that tetrameric crosslinkers can undergo. Each of the four actin-biding domains in the tetrameric crosslinker binds actin with rate $k_{bind}$ and unbinds in a force-sensitive manner with unbinding rate $k_{unbind}$. **E** Schematic depicting the

various actin-binding and protein-protein binding reactions that monomeric units in the dynamic multimerization model can undergo. Each monomeric unit has a single actin-binding site and two protein-protein binding sites to facilitate the formation of large multimers. For each protein-protein binding domain, monomers bind to each other with a multimer formation rate $k_{form}$ and split in a force-sensitive manner with a multimer splitting rate $k_{split}$. **F** The geometry of the droplet is modeled by a dynamic ellipsoid with principal axes a, b, and c. **G** Droplet boundary deformation is regulated by the droplet surface tension $\sigma_{surface}$ and the effective viscosity $\mu_{effective}$. The deformation of the droplet is determined by the balance between the droplet surface tension $\sigma_{surface}$ and the point forces exerted by the filaments on the boundary. The time evolution of deformation is further modulated by $\mu_{effective}$, which acts as a damping parameter of the droplet shape dynamics.

not it is bound to an actin filament [B] or free [F], thus resulting in six different states that each monomeric unit can cycle between through a series of actin or monomer binding and unbinding reactions (Fig. 1E).

The space representing the condensate modeled the 3D geometry confining where reactions, filament dynamics, and diffusion occur, as in previous work[5,39,40]. Whereas our previous models used a spherical geometry that was rigid or deformed in a quasi-static manner[5,39,40], we now incorporate a deformable ellipsoid geometry that deforms dynamically in accordance with the forces acting on the boundary against the surface tension of the droplet (Fig. 1F, G). This framework in Cytosim was developed in Dmitrieff et al.[53] to understand the role of cortical tension in microtubule assembly. By modeling the condensate geometry as a dynamic ellipsoid, we investigated the feedback between condensate properties and the actin network. The extent of deformation is calculated by the balance between the interfacial surface tension and the point forces exerted by the filaments on the boundary−subject to volume incompressibility constraints[53]. Point forces are calculated from the force exerted to keep actin filaments within the boundary, as modulated by the boundary repulsion stiffness

(Supplementary Fig. 1). The speed at which deformation occurs is further modulated by an effective viscosity, $\mu_{effective}$, which acts as a damping parameter to slow the deformation of the ellipse axes over time[53].

Specific parameters used to set up the Cytosim model can be found in Supplementary Table 1, while the list of parameters varied in each set of simulations can be found in Supplementary Table 2. The Cytosim simulations were performed on the Triton Shared Computing Cluster (TSCC) at the San Diego Supercomputer Center (SDSC)[54].

Our investigations are organized as follows: we first show that in droplets containing VASP, allowing for droplet deformation leads to the emergence of discs (weakly-bundled actin structures) that are oriented along the axis of deformation. We found that a power law relationship exists between the initial radius of the droplet and the number of filaments for a particular aspect ratio. Our simulations reveal that the kinetics of droplet deformation follow a dynamic snapping behavior, a snap-through of the shape of the droplet that occurs over time, depending on the organization of the actin network in the VASP droplet. We then predicted that actin filament length plays

an important role in determining the onset and extent of droplet deformation; this prediction is verified by experimentally changing the capping protein concentration. We also show that all our observations are generalizable to ABPs that form dynamic multimers, establishing that actin bundling and droplet deformation do not require strong bundling proteins. Even in the absence of a specific ABP in the droplet, we show that actin filaments form discs that align along the axis of deformation, and therefore, concentrate actin in phase-separated containers is a sufficient condition for bundling and force generation. Finally, we show that the biochemical properties of actin crosslinkers drive actin structure formation by influencing the interplay between droplet shape and bundle alignment.

## Deformable VASP droplets produce tightly-bundled actin rings and weakly-bundled discs

To consider the importance of VASP droplet deformation in actin assembly, we relaxed the assumption of a rigid droplet interface used in our previous work, where we simulated actin bundling in rigid VASP droplets[5,39,40]. Here, we model simple deformation of a droplet, consistent with experiments where we observe droplet deformation[5,6,39,40], and ask whether the deformability of the droplet influences actin assembly. In our simulations, the droplet was allowed to deform with a given surface tension and a viscous dissipation parameter, along with a constant volume constraint[53]. To map the interaction between the droplet interface and the actin bundling, we simulated a range of binding and unbinding kinetics of VASP with actin filaments in deformable droplets (Fig. 2). In these deformable droplets, we observed the formation of weakly-bundled discs of actin filaments and rings; the weakly-bundled discs were seen in the regime of weak VASP binding kinetics ($k_{bind} \geq 0.1$ and $k_{bind} < k_{unbind}$ or $k_{bind} < 0.1$) (Fig. 2A). This is different from the case of rigid droplets, where we observe shells and rings (Supplementary Fig. 2A and Chandrasekaran et al.)[39]. The disc-shaped bundles that emerged in deformable droplets appeared to align themselves along the axes of maximum droplet deformation to minimize the energetic cost of filament bending (Fig. 2A). We quantified the thickness of the discs by analyzing the surface area fraction of the droplet covered by actin[39]. The differences in surface area fraction for rings and discs are shown in Fig. 2B, where rings form with a low surface area fraction, while discs have a much larger surface area fraction similar to shells seen in rigid droplets (Fig. 2B, Supplementary Fig. 2B). Additionally, when comparing the fraction of VASP tetramers bound to actin filaments, almost no VASP tetramers were bound to any actin filaments in simulations with kinetic conditions that favored disc formation (Fig. 2C). This is in contrast to rings and shells (Supplementary Fig. 2C), where kinetic trapping by VASP tetramers was required to tightly bundle actin filaments into rings or tightly confine the actin filaments within the original spherical geometry of the droplet so that the filament network develops into shell-like shapes[39].

Finally, we used principal component analysis (PCA) to classify the different actin network structures. PCA showed that for the data in Fig. 2B, C, which is governed by five independent parameters, the first three principal components (PCs) represent ~96% of the variance in the dataset (Supplementary Fig. 3A). To understand how each of the five parameters used to describe actin organization contributes to each PC, we then looked at the varimax loading parameter. We found that PC1 corresponded to information on the actin-covered surface area fraction and the fraction of VASP tetramers fully bound to four filaments, while PC2 captured information regarding the number of VASP tetramers bound to one and two filaments (Supplementary Fig. 3B). Then, using a K-means clustering algorithm on the first three PCs, we classified the resulting actin network shapes into four distinct clusters (Fig. 2D). We determined that four is the optimal number of clusters by choosing the largest silhouette coefficient among various cluster sizes (Supplementary Fig. 3C). The actin network shapes formed by each

combination of kinetic parameters are given in Supplementary Table 3. We computed the eigenvalues of the gyration tensor of the network to extract the ellipsoid spans (i.e., semi-axes where $a \geq b \geq c$). This analysis allowed us to approximate the shape of each actin network as an ellipsoid, with span a as the semi-major axis and span c as the semi-minor axis, that best described the distribution of filament positions. This analysis showed that the cluster with the lowest span c corresponded to rings, the cluster with an intermediate span c value corresponded to discs, and the two clusters with the highest span c corresponded to both loose and tight shells (Fig. 2E). This interpretation matched with a visual inspection, which revealed that these four clusters correspond to actin networks that form discs, rings, loose shells, and tight shells. Notably, the cluster corresponding to discs correlated positively with PC1 and negatively with PC2, meaning that the VASP crosslinkers in disc conditions were largely bound to very few to no filaments at all. The absence of a strong contribution from the crosslinkers suggests that disc formation was due to energy minimization of the interplay between droplet surface energy and filament bending energy rather than the result of VASP crosslinker-mediated kinetic trapping.

## Actin bundle thickness and droplet diameter satisfy a power-law relationship

We next sought to systematically understand the dynamics of droplet deformation. We first analyzed how the aspect ratio of the droplet evolved over time under disc-forming conditions (weak VASP binding kinetics, Fig. 2, $k_{bind} = 0.001 \, s^{-1}$) for a droplet of surface tension of $\sigma_{surface} = 7 \, pN/\mu m$. We observed that the aspect ratio evolved non-monotonically with a consistent time signature for all disc-forming conditions. Until the filament length reached the diameter of the droplet, the aspect ratio remained unchanged ($L_{fil} = 2R \, \mu m$, $t \approx 200 \, s$). Once the filaments grew longer than the droplet diameter, the droplet aspect ratio increased until the filament length reached $L_{fil} = \pi R \, \mu m$ ($t \approx 300 \, s$). As the filaments bent around the periphery of the droplet, the aspect ratio decreased, indicating a partial recovery of the spherical shape after an ellipsoidal deformation. However, when the bundles of actin grew longer than $4\pi R/3 \, \mu m$, the aspect ratio returned to growing monotonically (Fig. 3A, Supplementary Fig. 4).

From these dynamics, three interesting questions emerge: What is the relationship between droplet size, filament bending energy, and the extent of droplet deformation? What factors control the onset of droplet deformation? And what factors control the dynamics of droplet deformation? We systematically answer these questions below.

To understand the relationship between the droplet size and extent of droplet deformation, we recall that disc formation is governed by an interplay between droplet surface energy and filament bending energy[6,29]. Since the filament bending energy is proportional to the thickness of the filament bundle, varying the number of filaments within our simulations will modulate the accumulation of filament bending energy in the system. Varying the droplet size, on the other hand, will change the droplet surface energy, which is given by the product of the surface tension and the surface area of the droplet. A parameter sweep of the number of filaments versus the initial diameter of the droplet will thus allow us to study their effects on droplet deformation. To test this idea, we simulated actin bundling in deformable droplets of surface tension $\sigma_{surface} = [2, 4, 8] \, pN/\mu m$ that ranged in size from an initial diameter of 0.5–4.0 $\mu m$ and varied the number of filaments from 5 to 100 under fixed VASP tetramer concentration (~0.40 $\mu M$). The growth rate of each filament was adjusted so that the final filament length matched the circumference of the original droplet ($2\pi R \, \mu m$ or $\pi D \, \mu m$) after 600 s for each droplet size. Our choice of surface tension values ensured that we were operating in the deformable regime established in our 30 filament, 2 $\mu m$ diameter droplet simulations (Supplementary Fig. 4). We simulated 960 unique conditions (10 replicates each, parameters tested shown in

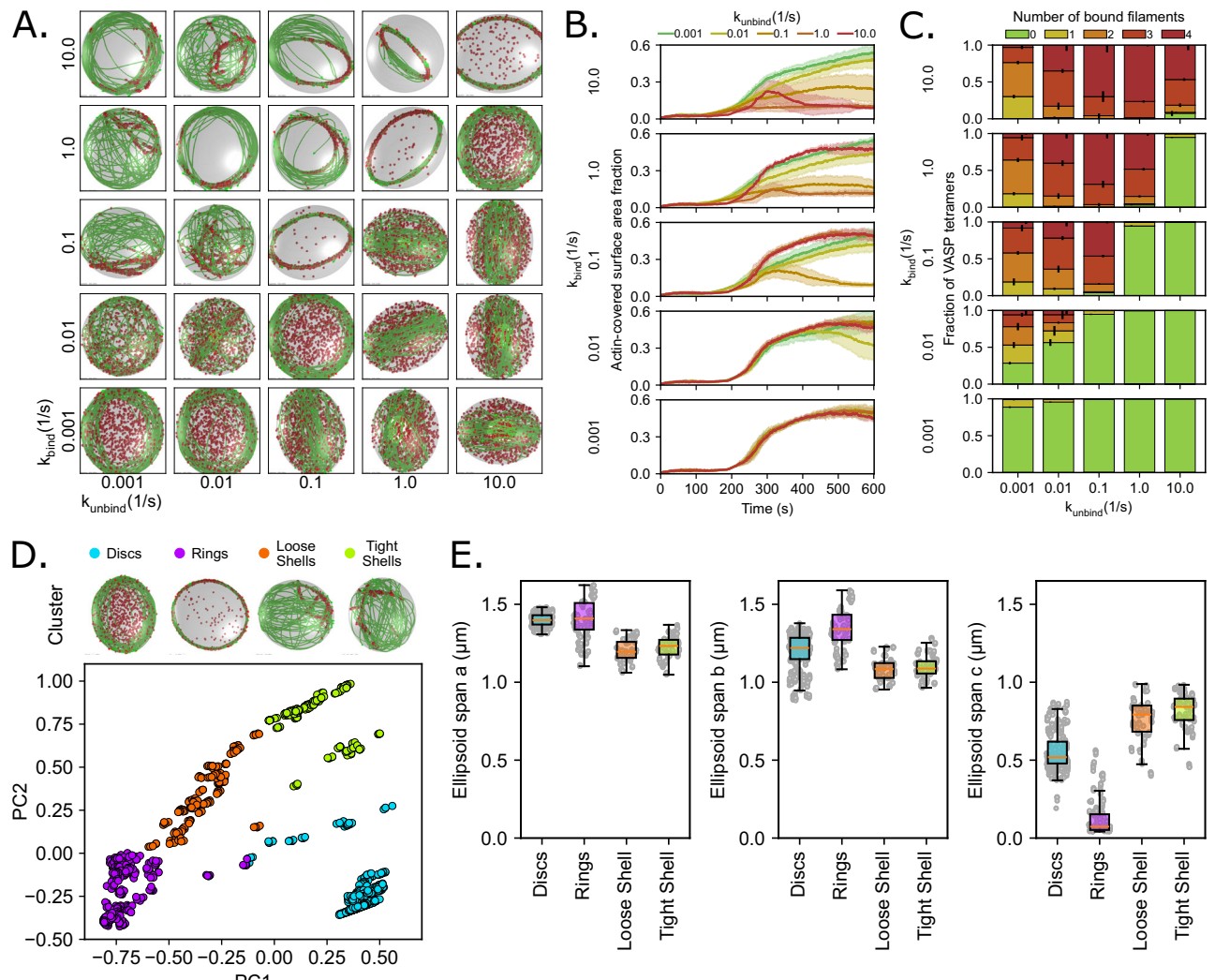

**Fig. 2 | Interactions between VASP binding kinetics and a deformable ellipsoidal droplet reveal the formation of weakly-bundled discs. A** Representative final snapshot ($t = 600$ s) from simulations at various binding and unbinding rates within droplets with a deformable ellipsoidal boundary (initially spherical with R = 1 μm) containing 30 actin filaments (green) and 1000 tetravalent crosslinkers (red spheres). The binding rates of the tetravalent crosslinkers are varied along each column, and unbinding rates are varied along each row. The polymerization rate at the plus (+) end is constant at 0.0103 μm/s, and neither end undergoes depolymerization. The deformable boundary has a surface tension of 7 pN/μm and an effective viscosity of 100 pN s/μm. **B** Time series showing the mean (solid line) and standard deviation (shaded area) of the actin-covered surface area fraction for varied tetravalent crosslinker binding and unbinding kinetics. **C** Fraction of tetravalent crosslinkers bound to 0, 1, 2, 3, or 4 actin filaments for each condition. The error bars represent the standard deviation. Data was obtained from the last 30 snapshots (5%) of each replicate. **D** K-means clustering was performed on the last five snapshots from each replicate (data shown in **B**, **C**), which revealed four cluster categories corresponding to actin structures that are discs, rings, loose shells, or tight shells. The data for each snapshot is projected onto a scatterplot of the first two principal components (PC1 and PC2), and colored by clusters. Representative snapshots of each cluster are included above the plot. Please see Supplementary Table 3 for the corresponding cluster identities for each combination of kinetic parameters. **E** Ellipsoid span a, span b, and span c (a ≥ b ≥ c) are calculated from the eigenvalues of the gyration tensor and approximate the shape of the actin network as an ellipsoid. The distribution of each ellipsoid span is depicted as box plots and organized by cluster category. Each boxplot displays the data points (gray), median (orange line), interquartile range (box), and 95% confidence interval (whiskers). Data was obtained from the last five snapshots of each replicate and sorted according to the cluster membership determined in (**D**): Discs ($n = 590$), Rings ($n = 310$), Loose Shells ($n = 165$), and Tight Shells ($n = 185$). For (**B**–**E**), 10 replicates are considered per condition. Source data are provided as a Source data file.

Supplementary Table 2). For each case with a final aspect ratio between 1.5 and 2.0, we recorded the number of filaments as an indicator of the requisite number of filaments to produce a small droplet deformation. This range was held constant across all 960 conditions (surface tension values, droplet radii, and number of filaments) studied to provide a common threshold with a substantial sample size.

We first investigated the case of VASP crosslinkers with disc-forming kinetics ($k_{bind} = 0.1\,s^{-1}$, $k_{unbind} = 1.0\,s^{-1}$) and found that the droplet aspect ratio increased with the number of filaments. Furthermore, as the initial diameter of the droplet increased, more filaments

were required to deform the droplet to a similar aspect ratio. We found that the number of filaments required to induce small deformations scaled with the initial droplet diameter according to a power law of the form $y = ax^b$, for all surveyed surface tensions for VASP crosslinkers with disc-forming kinetics (Fig. 3B). This power law relationship held even under ring-forming conditions (Fig. 3D).

This observation aligns with experimental findings that were shown in our previous work, wherein the thickness of the bundles that deformed actin rings to higher aspect ratios increased with droplet diameter[6]. VASP droplets that were deformed into higher aspect ratio

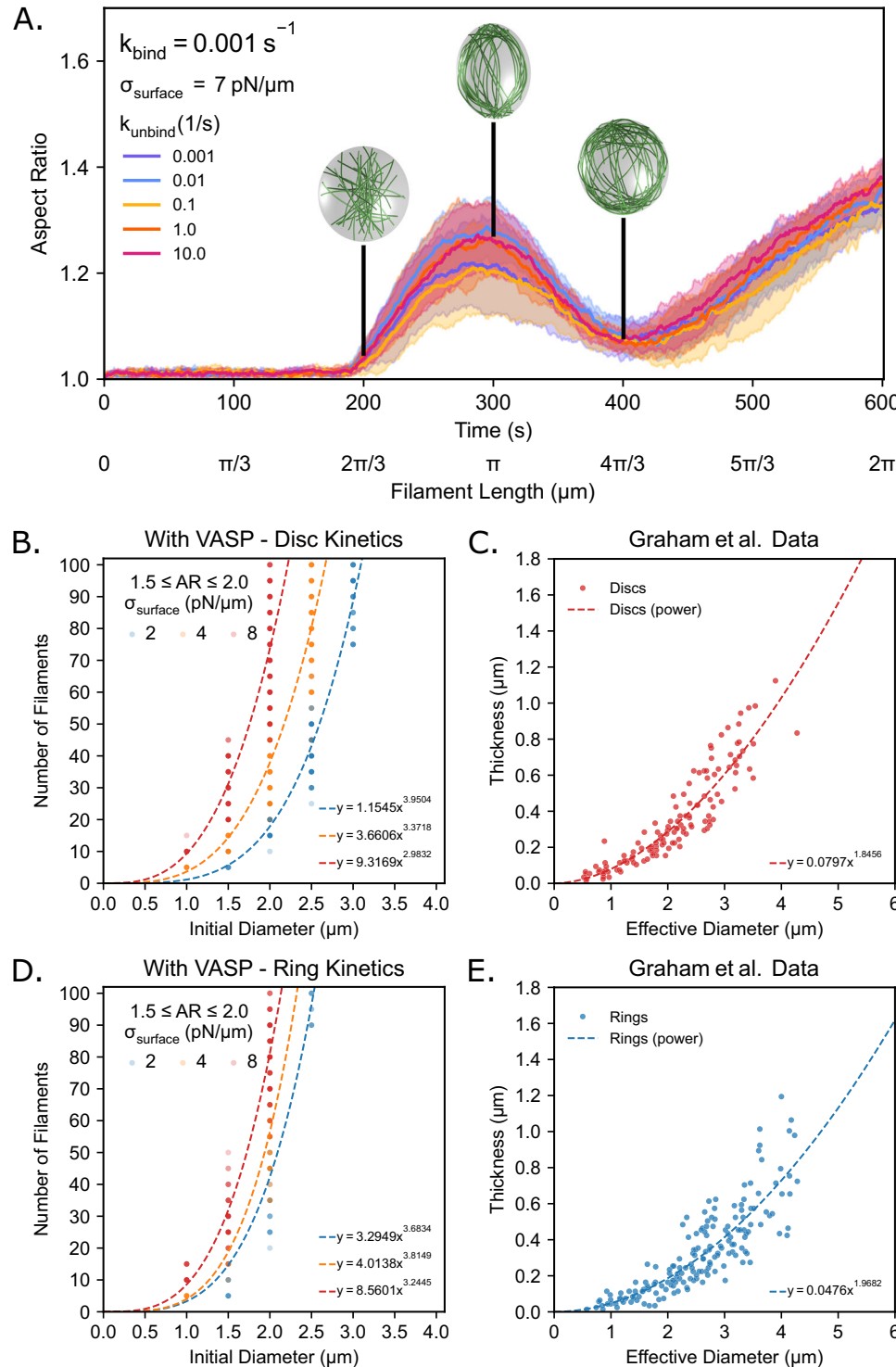

**Fig. 3 | Power law scaling relationship between the initial diameter of the droplet and the number of filaments. A** Time series showing the mean (solid line) and standard deviation (shaded area) of droplet aspect ratio for each $k_{unbind}$ condition when $k_{bind} = 0.001\,s^{-1}$. The aspect ratio is defined as the ratio between the longest and shortest axes of the ellipsoid (AR = a/c), where a ≥ b ≥ c. The black bars at timestamps $t = [200, 300, 400\,s]$ correspond to when each filament is roughly of length $[2\pi/3\,\mu m, \pi\,\mu m, and\,4\pi/3\,\mu m]$, respectively. Inset images are visualizations of the filaments and boundary without rendering crosslinkers. **B** Power law relationship for simulations with VASP that have disc-forming kinetics and with a surface tension of 2 pN/μm ($R^2 = 0.9404$, $n = 131$), 4 pN/μm ($R^2 = 0.8723$, $n = 179$), and 8 pN/μm ($R^2 = 0.7513$, $n = 170$). Data for each power law fit is taken from final droplet

aspect ratios at $t = 600\,s$ that are between 1.5 and 2.0. **C** Actin ring thickness as a function of the effective droplet diameter for condensates with aspect ratios <1.1 ($R^2 = 0.8430$, $n = 131$). Data were taken and reanalyzed from Fig. 3E of Graham et al.[6]. **D** Power law relationship for simulations with VASP that have ring-forming kinetics and with a surface tension of 2 pN/μm ($R^2 = 0.8190$, $n = 51$), 4 pN/μm ($R^2 = 0.8201$, $n = 66$), and 8 pN/μm ($R^2 = 0.9038$, $n = 116$). Data for each power law fit is taken from final droplet aspect ratios at $t = 600\,s$ that are between 1.5 and 2.0. For (**A, B, D**), 10 replicates are considered per condition. **E** Actin ring thickness as a function of the effective droplet diameter for condensates with aspect ratios <1.1 ($R^2 = 0.7278$, $n = 173$). Data were taken and reanalyzed from Fig. 3E of Graham et al.[6]. Source data are provided as a Source data file.

structures tended to have thicker actin rings in comparison to droplets that remained spherical. In these experiments, a power law relationship was also seen between actin ring thickness and the effective diameter of the VASP droplets. We reanalyzed the experimental data from those prior experiments and compared them to the model predictions in this study (Fig. 3C, E). These experimental results corroborated the computational predictions, and together the model and experimental findings suggest that the thickness of the actin ring contributes to the deformation of the VASP droplets and follows a power law growth relationship with droplet diameter. We note that the existence of a power law relationship is also consistent with previous theoretical predictions[29] and energetic arguments[6]. Differences between the power law exponents in the simulations and those predicted from theory arose from the aspect ratio range chosen for the simulations. The derivation of this power law relationship can be found in the Methods. Thus, using our agent-based simulations, we were able to confirm the existence of a power law relationship between the initial diameter of the droplet and the number of actin filaments for different droplet surface tensions.

## Filament length determines the onset and extent of droplet deformation

We next investigated the role of filament length on droplet deformation. In our prior studies, we showed that the filament length controls both the onset of ring formation and the final aspect ratios as the rings deform the droplet[6,39]. However, the effect of the dynamics of nondeterministic actin elongation on ring formation remained unknown. Therefore, we investigated the relationship between filament length and droplet deformation under ring-forming conditions in the presence of actin capping protein, which arrests polymerization of filaments[55]. Specifically, we simulated the stochastic capping and uncapping of actin filaments (Supplementary Fig. 5) using an implicit capping and uncapping framework. In our simulations, plus ends were allowed to switch between capped (at rate $k_{cap}$) and uncapped (at rate $k_{uncap}$) states stochastically throughout the simulation. Once capped, the end could not grow until the end was uncapped. As a result, the filaments reached a distribution of lengths and were crosslinked by VASP tetramers to form rings ($k_{bind} = 10.0\,\mathrm{s}^{-1}$, $k_{unbind} = 1.0\,\mathrm{s}^{-1}$) within both rigid (Supplementary Fig. 5A–C) and deformable droplets (Fig. 4). In this implicit framework, while we specified the rate of filament (un) capping, we did not specify a total number of capping protein molecules.

We varied the capping rates in the range [$k_{cap} = 1.0\,\mathrm{s}^{-1}$–$11.314\,\mathrm{s}^{-1}$] for fixed uncapping rate $k_{uncap} = 1.0\,\mathrm{s}^{-1}$ (Fig. 4A). These values were chosen because they produced the full range of VASP-crosslinked actin structures from linear bundles to rings (Supplementary Fig. 5A). We found that increasing capping rate ($k_{cap}$) resulted in shorter filaments and a narrower distribution of filament lengths (Fig. 4B). Additionally, decreasing capping rates in the range [$k_{cap} = 0.088\,\mathrm{s}^{-1}$–$1.0\,\mathrm{s}^{-1}$] also resulted in a narrower filament length distribution but centered around a larger filament length (Supplementary Fig. 5D). We then quantified the final aspect ratio of the droplet as a measure of the extent of the deformation. Increasing the capping kinetic rate resulted in lower aspect ratio droplets because shorter filaments cannot exert large deforming forces on the droplet surface (Fig. 4C). From these simulations, we predicted that when capping protein was added to VASP droplets with actin in them, the average droplet aspect ratio would decrease, and there would be fewer droplets with high aspect ratios.

To test this prediction in experiments, we incorporated capping protein, CapZ, into droplets composed of VASP. CapZ is an actin-binding protein that regulates filament dynamics by binding tightly to the barbed ends of actin filaments, thereby preventing further monomer addition and elongation and effectively controlling filament length[56–58]. Specifically, we formed droplets in a 10 μM solution of VASP

in the presence of 3 μM actin with increasing concentrations of CapZ. Droplets were formed by mixing VASP and associated proteins with 3% (w/v) PEG 8000 as a crowding agent. PEG is often added in the study of LLPS to mimic the crowded environment in the cell cytoplasm[59–61]. As observed previously[6,39,40], VASP droplets lacking CapZ deformed into rod-like shapes upon actin polymerization (Fig. 4D). As increasing concentrations of CapZ were added, the droplets became more spherical (Fig. 4D), and both the fraction of deformed condensates, defined as protein droplets with an aspect ratio greater than 1.2, and the average aspect ratio of droplets decreased. (Fig. 4E, F). Notably, phalloidin-stained actin did not arrange into rings at the inner surfaces of droplets in the presence of CapZ (Supplementary Fig. 5E), suggesting that CapZ addition reduced actin filament length such that there were no filaments long enough to exceed the condensate diameter.

## The dynamics of droplet deformation exhibit a dynamic snapping behavior that depends on the droplet surface tension

We previously showed that the dynamics of droplet deformation for deformable VASP droplets can be non-monotonic in time (Fig. 3A). To investigate how these dynamics depend on VASP and droplet properties, we tracked the droplet deformation dynamics for different VASP binding/unbinding rates and for different values of droplet surface tension ($\sigma_{surface} = 7\,\mathrm{pN/\mu m}$ (Fig. 5A–C) and $\sigma_{surface} = 2\,\mathrm{pN/\mu m}$ (Fig. 5D–F)). Droplet deformation can be understood as the balance between the bending energy of the actin bundle and the surface energy of the droplet. The onset of droplet deformation occurs when the length of the bundle exceeds the diameter of the droplet because the linear actin bundles are able to overcome the surface energy of the droplet. However, the extent of droplet deformation depends on VASP-actin interaction kinetics. When $k_{bind} > k_{unbind}$, actin in ring-like states with strongly-bound VASP tetramers leads to minimal droplet deformation. These rings are kinetically trapped, which results in reduced deformation capacity of the actin networks (Fig. 5B, C, Supplementary Movie 1).

When $k_{bind} \leq k_{unbind}$, we observed an increase in droplet deformation depending on the value of the surface tension. Under disc-forming conditions, where actin bundling is weak, the droplet begins to deform once the filaments reach $L_{fil} > 2R$ ($t \approx 200\,\mathrm{s}$). As the filaments grow longer, the filaments begin deforming the droplet linearly with time for low surface tension ($\sigma_{surface} = 2\,\mathrm{pN/\mu m}$) to form rod-like bundles of actin (Fig. 5D, E, Supplementary Movie 2). However, the variability increased with time in low surface tension and low binding and unbinding rates. In this case, we observed that out of ten replicates, six of them were rods while four of them snapped back into discs (Fig. 5D, left-side snapshots). On the other hand, when $\sigma_{surface} = 7\,\mathrm{pN/\mu m}$, the droplet aspect ratios evolve as a non-monotonic hump ($t \approx 200$–$400\,\mathrm{s}$), after which they either experience continuous linear growth or remain stable. This nonlinear temporal behavior can be understood as follows: the aspect ratio of the droplet increases initially with filament growth as filaments are aligned along the ellipsoidal major axis (Supplementary Movie 3). This anisotropic filament distribution enables filament growth-driven droplet deformation. As the filament length reaches $L_{fil} = \pi R\,\mu m$ at $t \approx 300\,\mathrm{s}$, the filaments bend along the periphery of the droplet, resulting in spindle-shaped networks that trace the contour of the ellipsoidal droplet, forming a near-isotropic network (Supplementary Movie 3). In this state, actin does not drive droplet deformation as the filament segments are arranged isotropically along the droplet surface. The actin filaments align themselves and grow along the droplet surface; such tangential growth does not lead to substantial forces exerted by the actin bundle on the droplet interface. As a result, the droplet surface energy overcomes the bending energy, and the droplet tends towards a spherical state. Eventually ($t > 400\,\mathrm{s}$, $L_{fil} = 4\pi R/3\,\mu m$), the filaments rearrange into discs that can overcome the surface energy, resulting in droplet

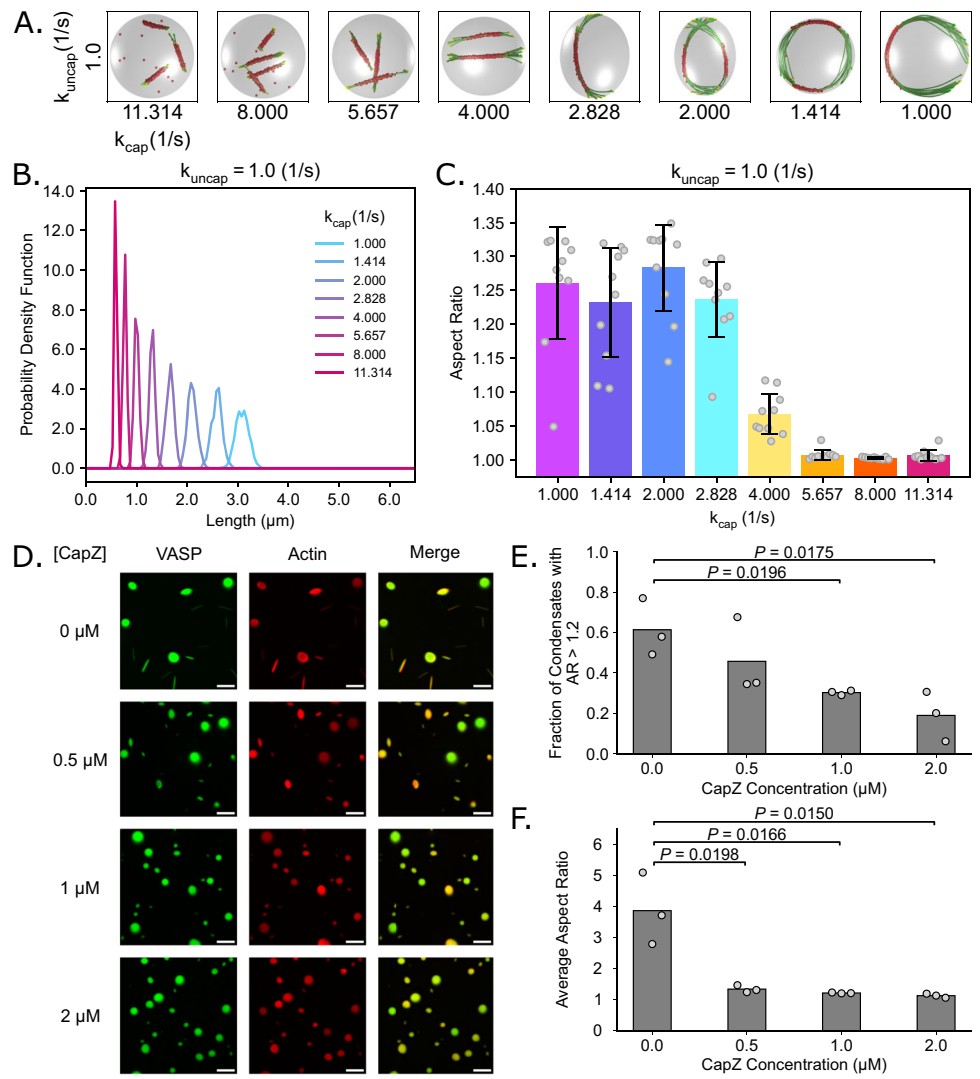

**Fig. 4 | Capping protein tunes filament length and, therefore, the extent of droplet deformation. A** Representative final snapshot ($t$ = 600 s) from simulations at various capping and uncapping rates within droplets with a deformable ellipsoidal boundary (initially spherical with R = 1 μm) containing 30 actin filaments (green) and 1000 tetravalent crosslinkers (red spheres). The filament growth rate is fixed at 0.0103 μm/s, and the capping rates are varied while the uncapping rate is held constant at 1.0 s$^{-1}$ to sample the transition between short rods and ring structures. The binding rates of the tetravalent crosslinkers are chosen to promote ring formation when $L_{fil}$ = 2πR μm. The deformable boundary has a surface tension of 4 pN/μm and an effective viscosity of 100 pN s/μm. **B** Probability density function of the final filament length for each simulation condition associated with A. As the capping rate increases, the probability of finding longer filaments decreases. **C** Bar chart showing the mean final droplet aspect ratio for each condition. The aspect ratio is defined as the ratio between the longest and shortest axes of the ellipsoid (AR = a/c), where a ≥ b ≥ c. The error bars represent the standard deviation, $n$ = 10.

For (**A**–**C**), 10 replicates are considered per condition. **D** Fluorescence images of condensates (3 μM Atto 594 labeled actin and 10 μM Atto 488 labeled VASP) with increasing CapZ concentrations [0 μM, 0.5 μM, 1 μM, and 2 μM] showing the suppression of high aspect ratio condensates as CapZ concentration increases. Scale bars, 5 μm. **E** Quantification of the fraction of condensates with an aspect ratio greater than 1.2 for the conditions shown in (**D**). Data are the mean across three independent experiments. The overlaid gray circles denote the means of each replicate. Significance values are determined using an unpaired, two-tailed t-test on the means of the replicates, $n$ = 3. **F** Quantification of the average aspect ratio for conditions shown in (**D**), showing a decrease in aspect ratio as CapZ concentration increases. Data are the mean across three independent experiments. The overlaid gray circles denote the means of each replicate. Significance values are determined using an unpaired, two-tailed t-test on the means of the replicates, $n$ = 3. Buffer conditions for all conditions were 20 mM Tris pH 7.4, 150 mM NaCl, 5 mM TCEP, and 3% (w/v) PEG 8000. Source data are provided as a Source data file.

deformation towards an ellipsoidal shape. This can be observed through the reduction in actin-covered surface area from ~300 to 400 s as shown in Fig. 2B. From ~200 to 400 s, the initial increase and decrease in aspect ratio forms a hump-like structure in the graph that can be understood in the context of a dynamic snapping behavior. Snap-through bifurcations are a common feature in mechanics when small changes to a control parameter can lead the system from one stable energy-minimizing state to another[62–66]. Albeit, in our case, we observe a dynamic snapping because the kinetic rearrangement of the actin filaments over time leads to a time-dependent response.

In the case of ring formation kinetics, the stronger actin binding-unbinding kinetics, specifically when $k_{bind} \approx k_{unbind}$ for $k_{bind} \geq 0.1$ (Fig. 5B, C), resulted in actin bundles that have formed by the time that the filament length reaches the diameter of the droplet, $L_{fil}$ = 2R μm ($t \approx 200$ s). By forming a bundle early, nascent ring structures emerge before $L_{fil}$ = πR μm ($t \approx 300$ s) in the form of either a single main bundle or two primary bundles oriented 180° from each other. Here, the filaments are arranged into the bundle to minimize bending energy and, with the guidance of crosslinkers that favor bringing filaments together, are better able to concentrate force to deform the droplet.

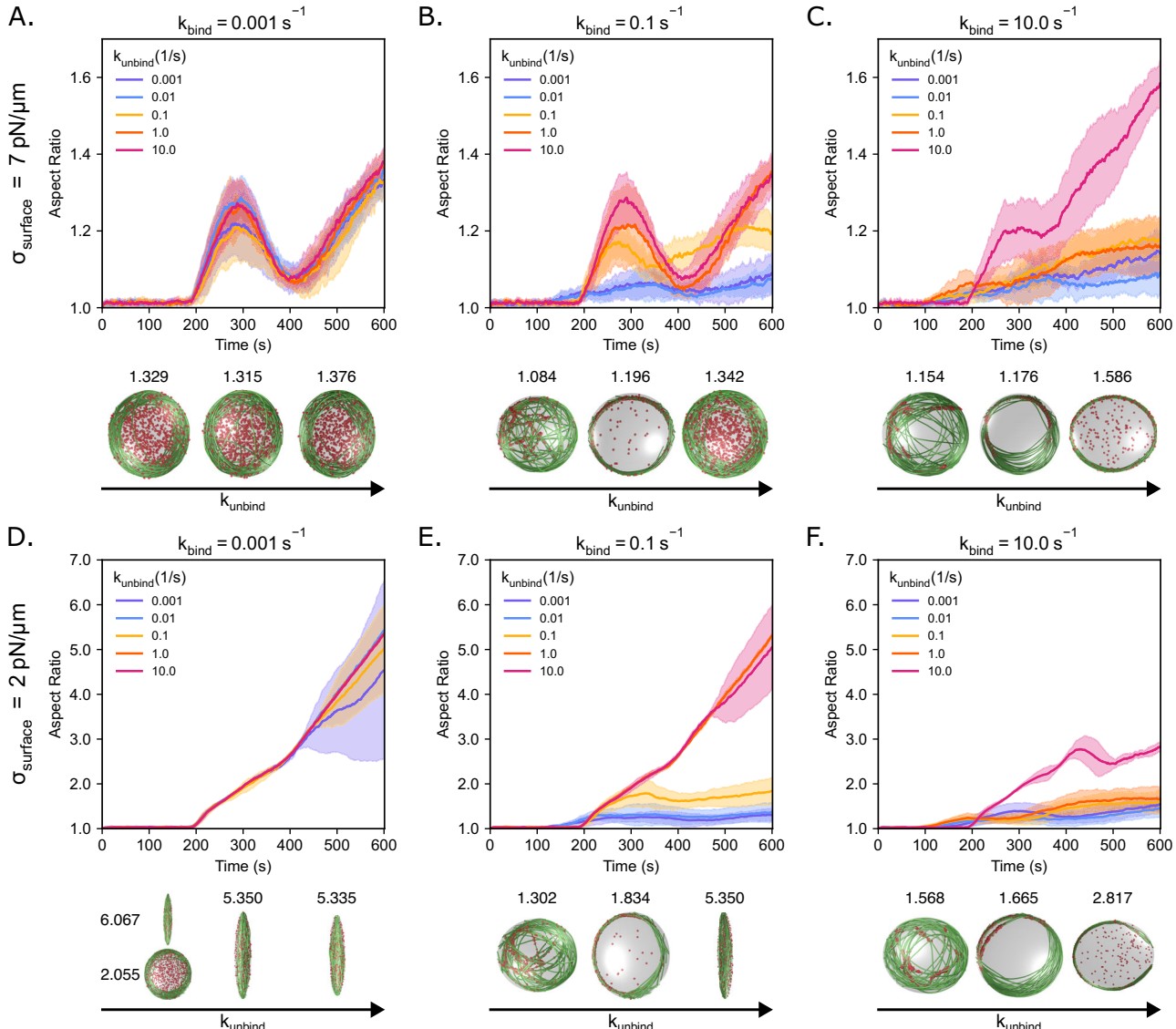

**Fig. 5 | VASP crosslinker binding and unbinding kinetics alter the dynamics of droplet deformation governed by the competition between filament bending energy and droplet surface energy.** Time series showing the mean (solid line) and standard deviation (shaded area) of droplet aspect ratio for each condition. The aspect ratio is defined as the ratio between the longest and shortest axes of the ellipsoid (AR = a/c), where a ≥ b ≥ c. For surface tension $\sigma_{surface}$ = 7 pN/μm: **A** $k_{bind}$ = 0.001 s$^{-1}$, **B** $k_{bind}$ = 0.1 s$^{-1}$, and **C** $k_{bind}$ = 10.0 s$^{-1}$. For surface tension $\sigma_{surface}$ = 2 pN/μm: **D** $k_{bind}$ = 0.001 s$^{-1}$, **E** $k_{bind}$ = 0.1 s$^{-1}$, and **F** $k_{bind}$ = 10.0 s$^{-1}$. Note that the aspect ratio scales are different between **A**–**C** and **D**–**F**. The snapshots show actin in green and VASP in red. High aspect ratio droplets have almost linear rod-like actin bundles. The snapshots within each panel correspond to $k_{unbind}$ values 0.001 s$^{-1}$, 0.1 s$^{-1}$, and 10.0 s$^{-1}$. Final aspect ratios are mentioned above each of the snapshots. For (**A**–**F**), 10 replicates are considered per condition. Source data are provided as a Source data file.

Crucially, because the filaments have generally been confined into a highly-aligned, essentially 2D ring structure stabilized by crosslinkers, the bending energy of the ring is able to overcome the droplet surface energy to push the droplet toward higher aspect ratios without exhibiting an initial non-monotonic growth pattern as was the case for discs (compare rings in Fig. 5C and discs in Fig. 5A).

**Deformable droplets made of ABPs that form dynamic multimers also exhibit actin bundling and dynamic snapping**

We next investigated whether the relationship between droplet deformation and actin bundling was specific to VASP alone or generalizable to other ABPs. To do so, we investigated the role of monomeric ABPs that have homomeric interfaces. Through the formation of multimers, the actin-binding valency of the complex was increased, enabling crosslinking. We simulated ABPs as monomeric units capable of binding to a single actin filament and up to two other monomers,

thus allowing for the formation of higher-order multimers (Fig. 1E)[5]. We prescribed ring-forming actin-binding kinetics, while modulating the kinetics of multimer formation and splitting, and found that actin rings and discs formed depending on the different kinetic parameters (Fig. 6A, Supplementary Fig. 6A). Our analysis of multimer lengths showed that rings only formed in kinetic conditions that favored multimerization of the crosslinkers, while discs formed under conditions in which the crosslinkers remained largely monomeric (Supplementary Fig. 6B, C).

We also examined the relationship between droplet size and the number of filaments. We chose the multimerization condition that maximizes multimer size (Supplementary Fig. 6C, $k_{form}$ = 10.0 s$^{-1}$, $k_{split}$ = 0.01 s$^{-1}$). We found that the initial radius of the droplet and the number of filaments required to deform it also followed a power law relationship (Fig. 6B) similar to stable tetramers studied earlier (Fig. 3B, D). To test our model prediction that dynamic multimers of ABPs were

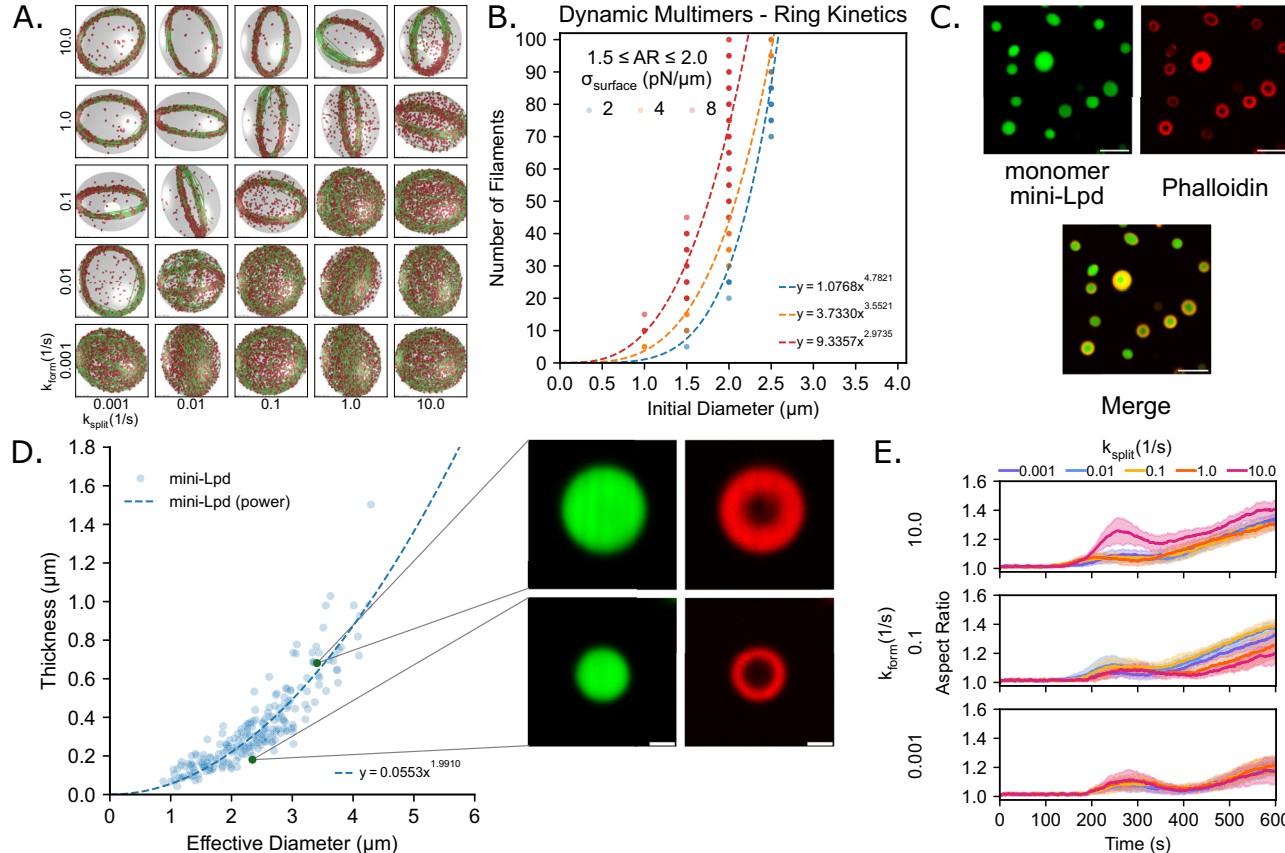

**Fig. 6 | Multivalent droplet environments exhibit disc formation and a power law scaling relationship between the initial diameter of the droplet and the number of filaments. A** Representative final snapshot ($t = 600$ s) from simulations at various multimer-forming and splitting rates within droplets with a deformable ellipsoidal boundary (initially spherical with R = 1 μm) containing 30 actin filaments (green) and 2000 monomeric crosslinkers (red spheres) that can dynamically form multimers of various lengths. The monomer-monomer binding rates of the tetravalent crosslinkers are varied along each column, and monomer-monomer unbinding rates are varied along each row. The actin-binding rates of the monomers are consistent with ring-forming kinetics determined from previous simulations. The polymerization rate at the plus (+) end is constant at 0.0103 μm/s, and neither end undergoes depolymerization. The deformable boundary has a surface tension of 7 pN/μm and an effective viscosity of 100 pN s/μm. **B** Power law scaling relationship between the number of filaments and the initial diameter of deformable droplets in dynamic multimerization simulations with a surface tension of

2 pN/μm ($R^2 = 0.9377$, $n = 103$), 4 pN/μm ($R^2 = 0.9440$, $n = 104$), and 8 pN/μm ($R^2 = 0.7472$, $n = 173$). **C** Representative confocal cross-section images of condensates formed from 5 μM monomer mini-Lpd after the addition of 3 μM actin and staining with Phalloidin iFluor 594, showing rings of polymerized actin. Buffer conditions were 20 mM Tris pH 7.4, 50 mM NaCl, 5 mM TCEP, and 3% (w/v) PEG 8000. Scale bars, 5 μm. **D** Power law scaling relation between the measured actin ring thickness and the diameter of monomer mini-Lpd condensates ($R^2 = 0.8251$, $n = 228$). The inset images correspond to the denoted points on the graph and depict examples of the difference in measured actin ring thickness. Scale bars, 1 μm. **E** Time series showing the mean (solid line) and standard deviation (shaded area) of droplet aspect ratio for each condition. The aspect ratio is defined as the ratio between the longest and shortest axes of the ellipsoid (AR = a/c), where a ≥ b ≥ c. For (**B**, **E**), 10 replicates are considered per condition. Source data are provided as a Source data file.

---

sufficient to bundle actin filaments and sustain droplet deformation within protein condensates, experiments were performed using condensates made of an EGFP-tagged C-terminal IDR region of the Lpd protein. This protein, henceforth called monomer mini-Lpd as it is a native monomer, was shown to phase separate and facilitate actin filament assembly[5]. To test if there was a power law relationship between actin ring thickness and condensate diameter, we performed similar experiments to those done previously with VASP (Fig. 3C, E)[6] with condensates composed of monomer mini-Lpd. Specifically, condensates were formed from a 5 μM solution of monomer mini-Lpd and 3 w/v% PEG 8000. Monomeric actin (G-actin) was added to the condensates at a concentration of 3 μM and allowed to assemble prior to filamentous actin staining with phalloidin-594 (Fig. 6C). Actin rings appeared at the droplet periphery. The thickness of the actin ring and the effective diameter of the encapsulating mini-Lpd condensate were quantified as described in the Methods. We then plotted ring thickness as a function of effective condensate diameter and extracted a power law scaling relationship (Fig. 6D). Ring thicknesses somewhat below

the optical diffraction limit were estimated by measuring the Gaussian intensity profile for actin at the droplet edge and subtracting the microscope point spread function. In line with model predictions, condensates formed from monomer mini-Lpd also displayed a power law relationship between condensate diameter and actin ring thickness, displaying similar values in the power law fit to those displayed in VASP condensates. Together, the experimental results and modeling predictions suggest that this power law relationship is generalizable to any condensate-forming protein that facilitates actin filament assembly.

In simulations, analysis of the deformation kinetics of these droplets showed that dynamic multimer crosslinkers promoted a higher final aspect ratio than the analogous condition for stable tetramers (VASP) (compare Fig. 6E and Fig. 5C, $k_{unbind} = 1.0$ s$^{-1}$). Furthermore, while all conditions experienced droplet deformation, the conditions that formed rings had larger final aspect ratios than the conditions that formed discs (Fig. 6E, Supplementary Fig. 6D). Thus, dynamic multimers form robust rings with additional modes of reorganization, which

facilitates the bending energy dissipation resulting in higher aspect ratios. This also alters the dynamics of droplet deformation, as reflected by the lack of a pronounced hump in the aspect ratio time series (Fig. 6E, Supplementary Fig. 6D).

### Confinement of growing actin filaments in liquid droplets is sufficient for filament bundling and droplet deformation in the absence of crosslinkers

Our analysis of VASP and Lpd droplets and their interactions with actin shows that weakly-bundled actin discs emerge when the droplet boundary is allowed to deform, even when crosslinker interactions with actin filaments are minimal. The lack of appreciable crosslinker activity suggests that disc formation might be primarily driven by the material properties of the droplet itself, rather than the crosslinkers. Therefore, we next investigated how droplet properties impacted actin bundling. Specifically, we varied $\sigma_{surface}$ and $\mu_{effective}$ in a deformable droplet in the absence of any crosslinking interactions. We found that actin filaments formed loose discs when the droplet surface tension was sufficiently low (Fig. 7A). The fraction of the droplet surface covered by actin was larger for shells than for discs, and the transition from discs to shells was clear as surface tension increased (Supplementary Fig. 7A, B). We also observed that shell structures were recovered as surface tension increased, indicating that droplet deformation could drive assembly of weakly-bundled actin discs. As expected, changes in the effective viscosity had a very muted effect on the final aspect ratio, as this parameter alters only the time necessary to deform the droplet and not the final state of deformation within the deformable ellipsoid framework[53]. Thus, we focused on the $\sigma_{surface}$ for the next steps.

We next investigated whether droplet deformation in the absence of crosslinkers also exhibits a power law relationship between the number of actin filaments and droplet diameter. As before, we varied the number of actin filaments and the initial diameter of the droplet and extracted a power law relationship for nonspecific crosslinkers (Fig. 7B). Again, we observed that actin elongation alone was sufficient to deform the droplet. This result led us to predict that an arbitrary condensate-forming protein with a weak affinity for actin filaments would be sufficient to form these discs. To experimentally test this prediction, we formed condensates from the RGG domain of LAF-1, a well-characterized phase-separating protein that lacks actin-binding domains[34], which has a high concentration of positively charged arginine residues and is therefore likely to interact non-specifically with negatively charged actin. We examined how the addition of actin impacted RGG condensate shape and filament organization. RGG droplets were formed from a 20 μM solution of the protein and 3 w/v% PEG 8000. Increasing concentrations of actin were added to the droplets after they formed, as described above. As predicted by the simulations, RGG condensates were progressively deformed into rod-like morphologies as the concentration of actin increased (Fig. 7C). Additionally, both the fraction of deformed condensates, defined as protein droplets with an aspect ratio greater than 1.2 (Fig. 7D), and the average aspect ratio of droplets increased with increasing actin concentration (Supplementary Fig. 7C). Notably, phalloidin staining revealed that actin filaments were arranged into rods and peripheral rings at the inner surfaces of the droplets (Supplementary Fig. 7D). These experimental observations suggested that actin filament assembly within condensates was alone sufficient to promote the bundling of actin filaments into discs, and that asymmetries in the condensate boundary could alone facilitate bundling of actin filaments into discs. We note that the RGG droplets were diffuse at larger condensate diameters. As ring thicknesses across a range of droplet diameters are necessary to derive a power law trend, we were unable to quantify the relationship between droplet diameter and ring thickness in RGG droplets. Finally, when we increased the surface tension in the simulations, we observed that the droplets deformed to lower aspect ratios. The dynamics of droplet deformation transitioned from dynamic snapping to a slow monotonic increase over time (Fig. 7E, Supplementary Fig. 7B).

### Feedback between droplet shape and bundle alignment is tuned by crosslinker properties

Throughout our simulations, we found different dynamics for droplet deformation depending on the crosslinker kinetics and multivalency and droplet surface tension (Figs. 3A, 4C, 5, 6E, 7E, Supplementary Figs. 4C, 6D, 7B). Based on these observations, we hypothesized that there exists a feedback between the droplet shape and filament alignment. To test this hypothesis, we first simulated actin filament elongation and bundling within rigid ellipsoidal droplets without any crosslinkers (Fig. 8A). We defined an alignment angle of the actin bundle as the angle formed between ellipsoid span a and span c, obtained from the eigenvalues of the gyration tensor as in Fig. 2E and described in the Methods. A filament alignment angle approaching 0° describes a flat distribution of actin, as expected for a ring, while an alignment angle approaching 45° describes a perfectly spherical distribution of actin, as expected for a spherical shell. We found a strong relationship between the droplet aspect ratio and filament angle (Fig. 8). To understand the role of boundary shape and actin organization, we simulated actin networks within rigid ellipsoidal droplets fixed at various aspect ratios (Fig. 8A). We found that the actin filaments aligned with the major axis of the ellipsoid at all aspect ratios studied (Fig. 8A). When we repeated these calculations with deformable droplets without crosslinkers, we found that decreasing surface tension resulted in both higher aspect ratio droplets and lower filament alignment values (Fig. 8B). Further, we observed that small changes to alignment angle resulted in large changes to aspect ratios (Fig. 8B, inset). This observation further validates our hypothesis that the filament alignment angle controls the extent of droplet deformation. When we analyzed the relationship between filament alignment angle and aspect ratio in the presence of dynamic multimers under ring-forming conditions, we saw a shift towards more aligned filaments and higher aspect ratios as multimer formation became more favorable (Fig. 8C, D). When we used stable tetrameric VASP as a crosslinker, we saw that, depending on the binding kinetics, we could achieve the same alignment angle for a wider range of aspect ratios (Fig. 8E, F). Comparing time profiles of aspect ratio and alignment angle evolution between ring-forming dynamic multimeric (Fig. 8D) and stable, tetrameric (Fig. 8F) crosslinkers, we observed that the stability of a multimeric unit also tuned the maximum aspect ratio achieved during deformation. Thus, we concluded that, while the material properties of the droplet determine the extent of filament alignment and droplet deformation, the biochemical properties of ABPs controlled the dynamic emergence of different actin networks.

## Discussion

In this work, we use a combination of agent-based modeling and in vitro biophysical experiments to systematically dissect how the biochemical and physical properties of liquid-like droplets of ABPs modulate actin filament organization and droplet shape. We show that confinement within deformable boundaries promotes actin bundling to form either tight rings or weakly-bundled discs characterized by spatially restricted actin filaments. Consequently, the bending energy of the disc or the ring deforms droplets to form ellipsoids. We show that the extent of the droplet deformation depends on both the interfacial properties of the droplet and the nature of crosslinking of the actin network. Filaments in tightly-bundled actin rings are unable to unravel to generate large forces on the droplet interface[67]. Thus, kinetically trapped actin rings can only induce moderate droplet deformation. Additionally, multivalent crosslinkers with similar binding affinities can allow for actin reorganization, leading to higher aspect ratio droplets. On the other hand, actin filaments in weakly-

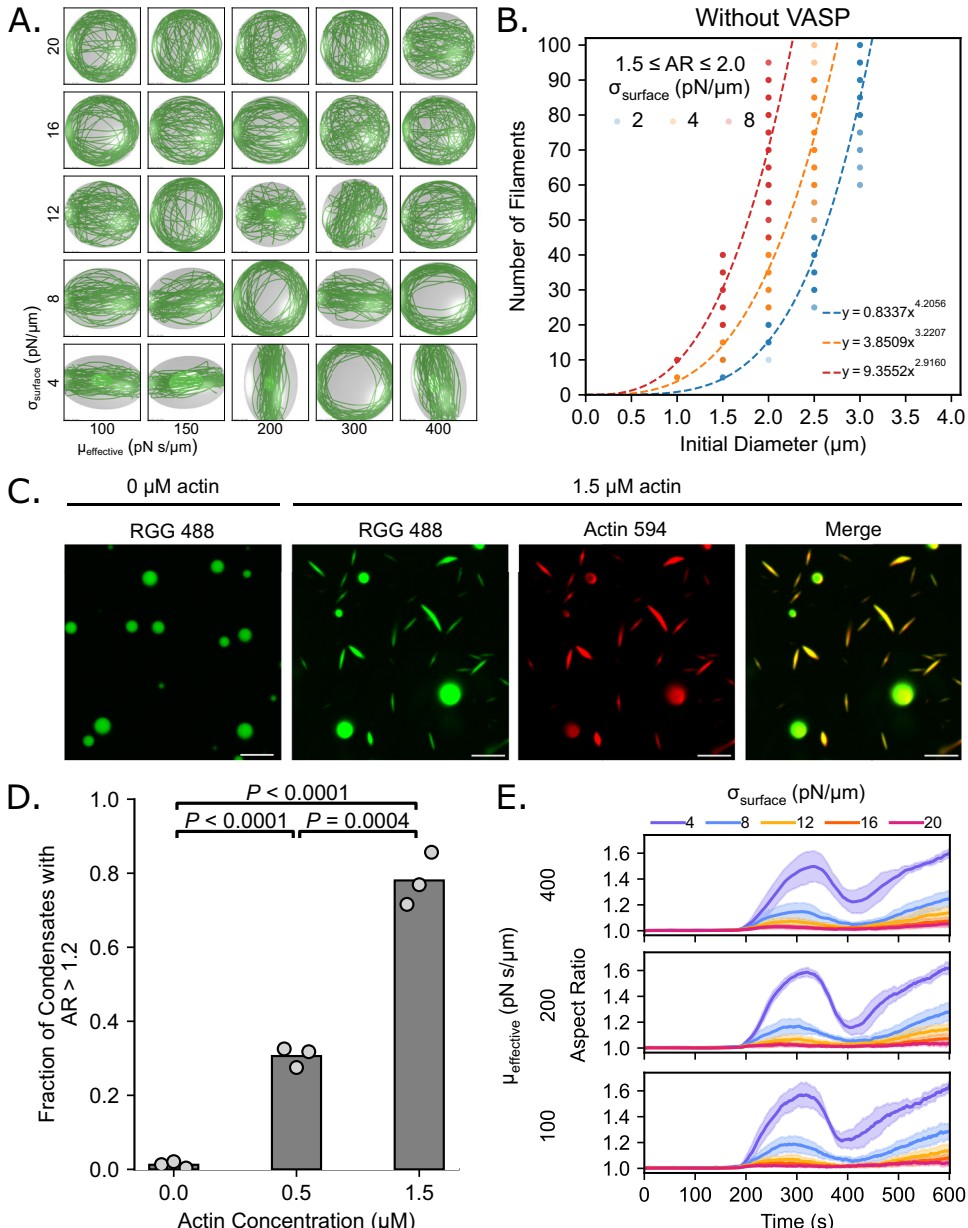

**Fig. 7 | Interfacial properties of the droplet surface govern deformation and determine whether the actin network will form shells or discs. A** Representative final snapshots ($t = 600$ s) from simulations varying the properties of the deformable droplet boundary (initially spherical with R = 1 μm) containing 30 actin filaments (green) and in the absence of any actin crosslinker. The effective viscosity, which attenuates the rate of deformation, is varied along each column, and the surface tension, which describes the innate resistance of the droplet boundary to deformation, is varied along each row. The polymerization rate at the plus (+) end is constant at 0.0103 μm/s, and neither end undergoes depolymerization. **B** Power law relationship for simulations without VASP present and with a surface tension of 2 pN/μm ($R^2 = 0.9230$, $n = 146$), 4 pN/μm ($R^2 = 0.8790$, $n = 165$), and 8 pN/μm ($R^2 = 0.7655$, $n = 155$). Data for each power law fit is taken from final droplet aspect ratios at $t = 600$ s that are between 1.5 and 2.0. **C** Fluorescence images of condensates (20 μM Atto 488 labeled RGG) with 0 μM and 1.5 μM Atto 594 labeled actin,

showing the lack of deformation in the absence of actin and high aspect ratio condensates in the presence of actin. Scale bars, 5 μm. **D** Quantification of the fraction of condensates with an aspect ratio greater than 1.2 for the conditions shown in C, showing an increase in aspect ratio fraction with increasing actin concentration. Data are the mean across three independent experiments. The overlaid white circles denote the means of each replicate. Significance values are determined using an unpaired, two-tailed t-test on the means of the replicates, $n = 3$. Buffer conditions for all conditions were 20 mM Tris pH 7.4, 50 mM NaCl, 5 mM TCEP, and 3% (w/v) PEG 8000. **E** Time series showing the mean (solid line) and standard deviation (shaded area) of droplet aspect ratio for each condition. Surface tension is varied within each plot with the same effective viscosity. The aspect ratio is defined as the ratio between the longest and shortest axis of the ellipsoid (AR = a/c), where $a \geq b \geq c$. For (**B**, **E**), 10 replicates are considered per condition. Source data are provided as a Source data file.

bundled discs can unravel into linear filaments, thereby maximizing droplet deformation to form rods. In both cases, the number of filaments needed to deform droplets scales as a power law with respect to droplet radius, indicating that larger droplets need more forces from actin filaments to deform. This general principle is true across all the cases tested in experiments and in simulations, and is also consistent

with experiments in other systems[29]. The prefactor of the power law depends on the interfacial properties of the droplet. Increasing the surface tension increases the prefactor, indicating that, for droplets with high surface tension, a higher number of filaments, and therefore a greater force, is needed to deform a droplet of the same diameter. These findings provide a mechanistic view of how

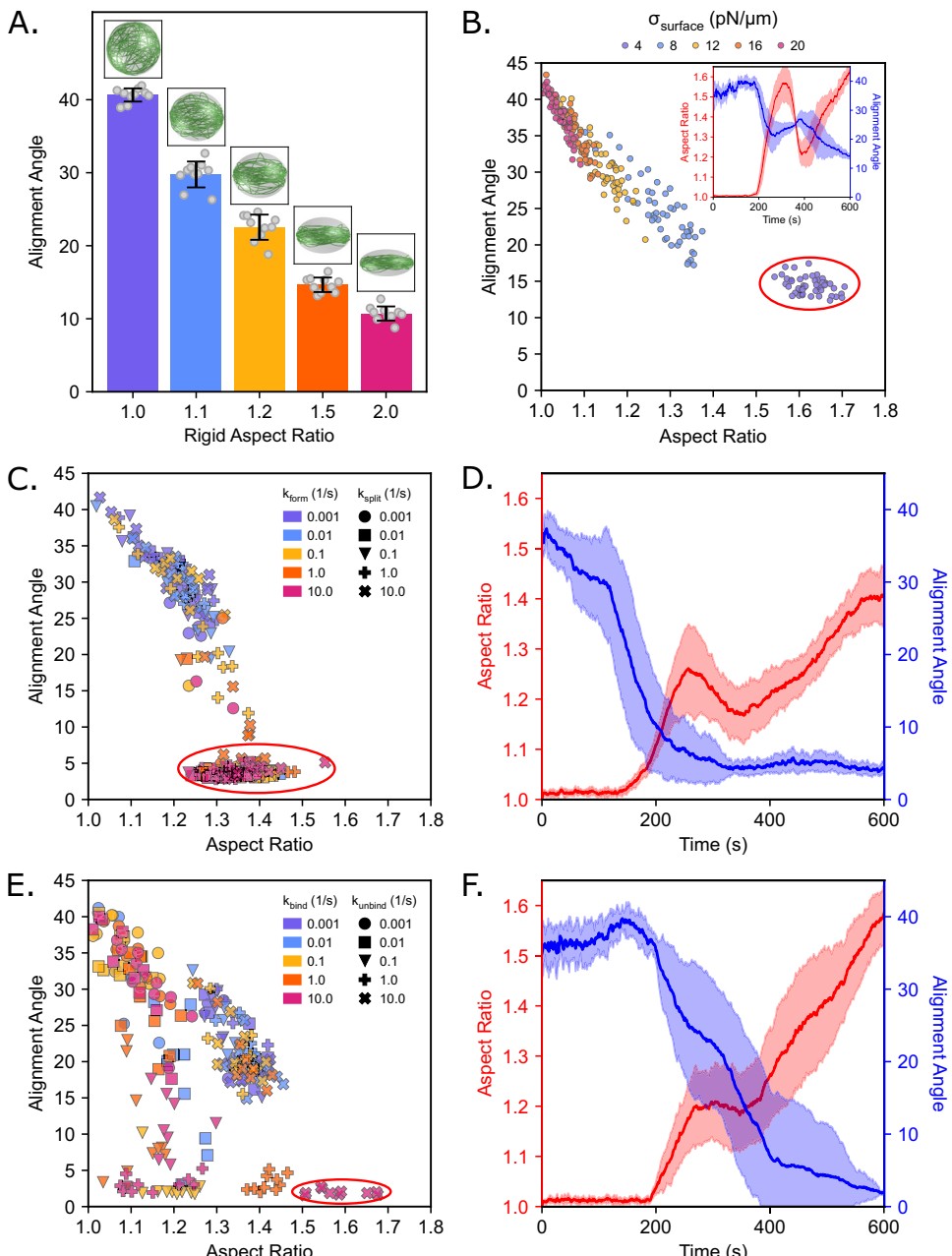

**Fig. 8 | Feedback between droplet deformation and actin filament alignment.**
**A** Mean final alignment angle for rigid ellipsoidal droplets with fixed but varied aspect ratios. The error bars represent the standard deviation, $n = 10$. Inset images are final representative snapshots of each condition. **B** Phase diagram plotting the final alignment angle and final aspect ratio across all replicates for simulations without crosslinkers. Each $\sigma_{surface}$ condition is indicated by color and include data for $\mu_{effective} = [100, 150, 200, 300, 400]$ pN s/μm. The inset graph shows the time series profile showing the mean (solid line) and standard deviation (shaded area) of the filament alignment angle (blue) and the droplet aspect ratio (red) for the condition indicated by a red circle ($\sigma_{surface} = 4$ pN/μm). **C** Phase diagram plotting the final alignment angle and final aspect ratio across all replicates for simulations with dynamically multimerizing crosslinkers. Each $k_{form}$ condition is indicated by color, and each $k_{split}$ condition is indicated by marker shape. **D** Time series profile showing the mean (solid line) and standard deviation (shaded area) of the filament alignment angle (blue) and the droplet aspect ratio (red) for the condition indicated by the red circle in (**C**) ($k_{form} = 10.0$ s$^{-1}$, $k_{split} = 10.0$ s$^{-1}$). **E** Phase diagram plotting the final alignment angle and final aspect ratio across all replicates for simulations with VASP crosslinkers. Each $k_{bind}$ condition is indicated by color, and each $k_{unbind}$ condition is indicated by marker shape. **F** Time series profile showing the mean (solid line) and standard deviation (shaded area) of the filament alignment angle (blue) and the droplet aspect ratio (red) for the condition indicated by the red circle in (**E**) ($k_{bind} = 10.0$ s$^{-1}$, $k_{unbind} = 10.0$ s$^{-1}$). For (**A**–**F**), 10 replicates are considered per condition. Source data are provided as a Source data file.

filament growth, bundling, and droplet deformability give rise to emergent properties of actin networks in liquid-like droplets and a holistic framework to understand previous experimental observations[5,6,12]. Thus, our work establishes the mechanochemical principles of droplet deformation as a function of the properties of droplets and actin networks.

Our finding that confinement alone is a bona fide mechanism of both filament bundling and droplet deformation has implications for how phase separation may organize the cytoskeleton. Previous studies have shown that actin confinement within rigid spheres[39,40] and inter-filament attraction are sufficient conditions for bundling[28,31]. Other studies have shown that confinement of actin in lipid bilayer vesicles

can also lead to filament organization[31,68,69]. Here, we highlight that even liquid deformable boundaries offer mechanical feedback to reorganize weakly-bundled filaments into discs. Additionally, consistent with previous results[28], we see that rings and discs organize to minimize bending energy within spheroidal droplets. More importantly, the feedback mechanism between droplet deformation and alignment of the actin bundle has implications for directional force generation of actin filaments in cells[19,70–72]. It is possible that specific ABPs, based on their kinetic interaction with actin, can lead to a wide range of bundling morphologies with distinct force-generation capacity. Such crosslinking dynamics also have implications for the mechanical properties of the actin networks[55,73,74].

Across the board, we found that the relationship between the crosslinker kinetics and droplet deformation has a rich array of dynamic behaviors that can be classified as: no change in aspect ratio, monotonic increase in aspect ratio, and, intriguingly, a non-monotonic dynamic behavior. While current imaging modalities were not able to capture the snapping behavior in experiments, the common principle driving droplet deformation dynamics across all cases we studied is that the droplet deforms when the energy exerted by the filament bundle overcomes the interfacial energy of the droplet. Viscosity regulates the rate of molecular diffusion, which affects the timescale of signaling and structure formation within condensates[3,5,24,34]. Surface tension regulates condensate shape and its interactions with other structures, such as actin filaments, through surface energy minimization[4,24,75–77]. This energy balance is consistent with what has been observed in other systems of deformable droplets[53]. Furthermore, the dynamics of droplet deformation mimic a dynamic snap-through, where at each time point, the balance of forces dictates the droplet shape, but the viscosity of the droplet sets a timescale of deformation[63,64]. Dynamic snap-through is observed in viscoelastic materials or when inertia is added to quasi-static equations of motion in mechanics[63,64]. In our case, we find that the kinetics of actin and crosslinker binding and unbinding add additional degrees of freedom, leading to the emergence of multiple endpoint morphologies. Thus, cytoskeletal polymers may leverage the availability of many ABPs to tune crosslinking and bundling kinetics to set the dynamics of force exertion.

Biomolecular condensates are emerging as a key regulatory mechanism of cellular phenomena. The idea that the containment of actin in a droplet can lead to feedback between bundling and condensate deformation can have implications in the organization of actin across a wide variety of biomolecular condensates. While we have shown such feedback in VASP condensates, these principles may be generalizable and apply to other condensates containing actin. For example, synapsin, a presynaptic protein that regulates synaptic vesicle clustering, has been shown to form condensates that recruit and organize actin along the condensate surface in a manner similar to the condensates described in this work[12,78]. Additionally, it has been recently shown in live cells that α-synuclein can regulate the material properties of synapsin droplets and alter their ability to cluster synaptic vesicles[79]. While many condensates form spherical three-dimensional droplets, recent studies have shown that two-dimensional condensates can form on the surface of the plasma membrane where transmembrane functions necessitate coupling between the condensate and the membrane. For condensates that assemble on surfaces, such as in focal adhesions and tight junctions, wetting and prewetting interactions with the membrane emerge from interfacial energy minimization, which depends on the material properties and characteristics of the condensate and surface[4,26,27,76,80–83]. These properties determine how the condensate spreads over the membrane, the droplet size, and the spatial patterning on the membrane[77,84,85]. The feedback between bundling and condensate deformation presented in this work may have implications for how surface-associated condensates respond to mechanical load, where shape changes and

interfacial geometry are important for force transmission. In focal adhesions, proteins such as talin and vinculin form dynamic and flexible two-dimensional condensate assemblies on the cellular membrane that cluster integrins and couple them to actin filaments, thus mediating the mechanical coupling between the actin cytoskeleton and the extracellular environment[7,8,82]. These assemblies undergo maturation under tension that is driven by actin filaments, which is accompanied by the reorganization of the actin architecture and changes in condensate shape, suggesting a mechanically analogous coupling between actin-generated forces and condensate deformation. Beyond confinement, in instances where the surface energy of the droplet is much smaller than the bending energy of the actin filaments, condensates can mediate actin crosslinking driven primarily by wetting filaments via capillary bridge interactions[24,75,86,87]. Additionally, N-WASP forms condensates on the plasma membrane that function as focus points for Arp2/3 activation. N-WASP condensates form as a result of a prewetting transition whereby condensation occurs once N-WASP reaches a critical surface concentration[81]. Together, these examples illustrate how feedback between condensate material properties, actin organization, and membrane properties may contribute to force generation and cytoskeletal remodeling at the membrane[88–90].

While our findings have broad implications for the interactions between the cytoskeleton and protein droplets, we acknowledge that we have constructed a simple model that reproduces the minimal physics of actin organization within condensates. To improve upon our current simulations, future models require the incorporation of hydrodynamic interactions between the actin filaments and the fluid droplet[91–94]. These hydrodynamic interactions may set the timescales for changes in the curvature of filaments and alter the bending energy of the actin bundles. Other studies involving large numbers of cytoskeletal filaments interacting with motor proteins have established the existence of swirling instabilities in cytoskeletal polymers[95,96]. We anticipate that the incorporation of additional physics in our framework has the potential to lead to a better understanding of actin responses in liquid-like droplets.

## Methods
### Model development
**Chemical and mechanical framework employed in Cytosim.** We constructed a minimal computational model to explore the emergent properties of actin networks in liquid droplets. Our simulations were performed in Cytosim (https://gitlab.com/f-nedelec/cytosim), an agent-based modeling framework that simulates the chemical dynamics and mechanical properties of cytoskeletal filament networks. Filament dynamics and diffusing species are modeled by numerically solving a constrained Langevin framework in a viscous medium at short time intervals. Additionally, actin filaments are composed of a series of inextensible rigid linear segments of length $0.1\,\mu m$ that are connected by hinge points to allow for bending. Cytosim computes the bending energy of the fiber using the specified flexural rigidity $k_{bend}$ in the input parameters. Tetrameric VASP crosslinkers are modeled as a single spherical solid of radius 30 nm with four actin-binding sites. Mini-Lpd monomers are modeled as spherical solids of radius 30 nm and function according to the dynamic multimerization model with a single actin-binding site and two monomer-monomer binding sites. Actin binding is governed by rate parameters $k_{bind}$ and $k_{unbind}$. Each binding site has a specified binding distance that specifies the radius of the spherical binding volume within which binding partners are considered as part of the binding reaction. Unbinding reactions are governed by Bell's law and Kramers' theory model representation of slip bond unbinding kinetics, where the rate constant is given by the unbinding rate and the characteristic force is given by the unbinding force. Steric repulsion potentials between diffusing crosslinkers and filaments are employed to avoid spatial overlap of species.

## Representation of actin elongation dynamics

Within the existing actin elongation framework, actin filaments elongate deterministically at a constant rate $k_{grow}$. Each filament is represented by a series of segments, each with a maximum length $L_{seg} = 0.1 \, \mu m$. Filament elongation rates are scaled to the size of the droplet space such that the filaments grow, if unperturbed, to the length of the circumference of the original droplet, $L_{fil} = 2\pi R \, \mu m$, by the end of the simulation at 600 s. As such, for a droplet with a radius $R = 1 \, \mu m$ where growth is deterministic, each filament will grow at a rate $k_{grow} = 0.0103 \, \mu m/s$ to reach $L_{fil} = 2\pi \, \mu m$ in length.

## Dynamics of capping protein

On top of the existing framework for filament elongation dynamics, the effect of capping protein is modeled implicitly by functionalizing the growing end of filaments with the ability to stochastically stop or restart elongation using a similar underlying framework as binding reactions in Cytosim. The capping reaction is governed by the rate parameters $k_{cap}$ and $k_{uncap}$. The implementation of our implicit capping protein model required edits to the Cytosim source code to simulate filament capping and uncapping, and has been made available on the GitHub repository alongside this publication.

## Underlying physics of dynamic ellipsoidal deformation model

The dynamically deformable ellipsoid framework for Cytosim was developed in Dmitrieff et al.[53] as a computationally expedient method for investigating the role of cortical tension in microtubule assembly within red blood cells. Whereas our previous models used a spherical geometry with a rigid boundary, we now incorporate a deformable ellipsoid geometry that dynamically adjusts the space and axes with each time step to model the condensate surface as a simple, continuously deformable surface. Deformation forces are calculated by the force balance between pseudoforces associated with the interfacial surface tension, $\sigma_{surface}$, the point forces exerted by the filaments on the boundary, and the pseudoforces associated with the pressure[53]. Pressure is represented as a Lagrange multiplier to the constant volume constraint used in this model. Time evolution casts the force balance along the three axes of the ellipsoid and calculates the speed at which deformation occurs as modulated by the effective viscosity, $\mu_{effective}$, which acts as a damping parameter to slow the deformation of the ellipse axes over time without affecting the final droplet shape[53]. Note that $\mu_{effective}$ (units pN s/$\mu m$) sets the timescale of droplet deformation and is a distinct parameter from the droplet viscosity (units pN s/$\mu m^2$), which we hold constant at $0.5 \, pN \, s/\mu m^2$ throughout all simulations. A full formulation of this model can be found in Dmitrieff et al. [53].

## Position evolution

Using the Langevin equation, a stochastic differential equation for describing Brownian motion, Cytosim is able to calculate the time evolution of 3D discretized points for the position of actin filaments and crosslinkers throughout the entire simulation space and duration of the simulation. For a system of N particles, there are 3N coordinates associated with the position of each particle $i$ in 3D space as given by $x_i = \{x_{i1}, x_{i2}, x_{i3}\}$. Each position $x_i$ is then evolved along each dimension $j$ as governed by the following representation of the Langevin equation:

$$dx^{ij}(t) = \mu f^{ij}_{tot}(t)dt + dB_j(t) \tag{1}$$

Here, $\mu$ is the viscosity of the solvent (not to be confused with $\mu_{effective}$). The first term on the right-hand side, $f^{ij}_{tot}(t)$, is the total force acting on each particle as a function of time. The second term on the right-hand side, $B_j(t)$, is the noise term for diffusion, which is a randomly sampled variable drawn from a normal distribution centered around a mean of 0 and with a standard deviation of $\sqrt{2D^i dt}$. The

diffusion constant $D$ is given by the Einstein relation $D = \mu k_B T$, where $\mu$ is the solvent viscosity, $k_B$ is the Boltzmann constant, and $T$ is the temperature.

## Dynamic multimerization model

We explore the role of multivalent interactions using the dynamic multimerization model, where independently diffusing monomers are simulated as solids that each have a single actin-binding site and two multimerization sites. These two multimerization sites allow each monomer to bind with up to two other monomers to dynamically form multimers such as dimers, trimers, tetramers, etc. This model introduces additional input parameters that describe the kinetics of multimer formation, $k_{form}$, and multimer splitting, $k_{split}$, and differs from the previous simulations, where static crosslinkers are prescribed as a single solid with multiple actin-binding sites. The monomer-monomer binding and unbinding reactions are modeled similarly to actin-crosslinker binding interactions already described. The implementation of our dynamic multimerization model required edits to the Cytosim source code to simulate multimerization reactions, as detailed in a previous publication[5].

## Simulations without specific crosslinkers

We explore the role of condensate deformation in directing filament organization in the absence of specific crosslinkers by constructing a model of actin polymerization within a liquid condensate where crosslinkers are omitted and only actin filaments are present within the droplet. This model serves as an analog to droplets composed of RGG, a protein that has only non-specific interactions with actin filaments.

## Theoretical scaling considerations for the droplet diameter and filament thickness

Here, we present the salient arguments laid out in Limozin et al.[29] and rederived in our earlier study[6]. Consider a liquid droplet of radius $R$ with an actin bundle of length $L$. The actin bundle is present below the interface between the droplet and the surrounding medium. We assume that the thermal fluctuations of the bundle dominate the droplet shape. The coordinate of the bundle centerline subject to fluctuations can be written as,

$$r(s) = R + u(s), \tag{2}$$

$$u(s) = Ru(\theta), s = R\theta, u(\theta) = \sum_m u_m \cos(m\theta). \tag{3}$$

Here, $s$ represents arc length and $r(s)$ represents the radial coordinate of the bundle centerline as a function of arc length. The radial coordinate is represented as fluctuations $u(s)$ around the droplet radius.

The bending energy of the bundle with flexural rigidity $k_b$ and local curvature $\kappa(s)$ is given by,

$$E_{bend} = \frac{k_b}{2} \int_0^L \kappa(s)^2 ds. \tag{4}$$

The flexural rigidity is related to the persistence length $L_p$ of the bundle by $k_b = L_p k_B T$, where $T$ is the temperature and $k_B$ is the Boltzmann constant. The local curvature can be written in polar coordinates as,

$$\kappa(s) = \sqrt{((x''(s))^2 + (y''(s))^2)}, \tag{5}$$

where

$$s = R\theta, \quad x = r \cos(\theta), \quad y = r \sin(\theta). \tag{6}$$

Substituting, we get,

$$\kappa(s)^2 = \left(r'' + \frac{r}{R^2}\right)^2 + \left(\frac{2r'}{R}\right)^2, \tag{7}$$

where, $r'(s) = \frac{1}{R}\sum m u_m \sin\left(\frac{ms}{R}\right)$, and $r''(s) = \frac{-1}{R^2}\sum m^2 u_m \cos\left(\frac{ms}{R}\right)$.

Now, we assume that the different modes (m with amplitude $u_m$) are uncorrelated, and ignore cross-correlations to write $\langle \kappa(s)^2 \rangle$,

$$\langle \kappa(s)^2 \rangle = \frac{1}{R^2} + \frac{2}{R^3}\sum_m u_m \langle \cos\left(\frac{ms}{R}\right)(1+m^2) \rangle + \frac{1}{R^4}\sum_m \left((1+m^4)u_m^2 \langle \cos^2\left(\frac{ms}{R}\right) + 2u_m^2 \sin^2\left(\frac{ms}{R}\right)\rangle\right) \tag{8}$$

Please note that the primary contribution to $\langle \kappa(s)^2 \rangle$ comes from the global curvature (droplet curvature) $\frac{1}{R}$ while the fluctuations contribute to higher order terms.

Substituting Eq. (8) in (4), we get,

$$E_{\text{bend}} = \frac{k_b}{2}\left(\frac{L}{R^2} + \frac{L}{R^4}\sum_m (2+m^4)u_m^2\right) \tag{9}$$

The primary contribution to the bending energy comes from the droplet radius. The fluctuation above that baseline energy $\Delta E_{\text{bend}}$ is given by,

$$\Delta E_{\text{bend}} = \sum_m \frac{k_b}{2}\frac{L}{R^4}u_m^2(2+m^4). \tag{10}$$

Applying the equipartition theorem, each mode contributes $k_B T/2$ energy, where $k_B$ is the Boltzmann constant and temperature $T$ as given by,

$$\frac{k_B T}{2} = \frac{k_b}{2}f(m)\frac{L}{R^4}u_m^2. \tag{11}$$

Here, we represent the mode-dependent terms by the function $f(m)$. Assuming that the ring thickness is proportional to the Fourier amplitude $u_m$, for a bundle of length $L = 2\pi R$, we get,

$$u_m \approx \frac{R^{1.5}}{\sqrt{L_p}} \tag{12}$$

Through this, we show that the bundle thickness has a power law relationship with droplet radius.

## Experimental methods
**Reagents.** Tris base, NaCl, Hellmanex III, Tris(2-carboxyethyl)phosphine (TCEP), poly-L-lysine, Atto 594 maleimide, and Atto 488 maleimide were purchased from Sigma-Aldrich. mPEG SVA was purchased from Laysan Bio. Phalloidin-iFluor594 was purchased from Abcam. Rabbit muscle actin was purchased from Cytoskeleton. Capping Protein (CapZ) was purchased from Hypermol.

**Plasmids.** A pET vector encoding the "cysteine light" variant of human VASP (pET-6xHis-TEV-KCK-VASP(CCC-SSA)) and monomer mini-Lpd (his-EGFP-Lpd(aa850-1250)) were gifts from Scott Hansen.

**Protein purification.** The pET-His-KCK-VASP(CCC-SSA) plasmid was transformed into *Escherichia coli* BL21(DE3) competent cells (NEB, Cat. no. C2527). Cells were grown at 30 °C to an optical density (OD) of 0.8. Protein expression was performed using the following steps, which were adapted from a method previously described but with minor alterations[5,6,41]. Expression of VASP was induced with 0.5 mM isopropylthiogalactoside (IPTG), and cells were shaken at 200 rpm at

12 °C for 24 h. The rest of the protocol was carried out at 4 °C. Cells were pelleted from 2 L cultures by centrifugation at 4785 g (5000 rpm in Beckman JLA-8.100) for 20 min. Cells were resuspended in 100 mL lysis buffer (50 mM sodium phosphate pH 8.0, 300 mM NaCl, 5% glycerol, 0.5 mM TCEP, 10 mM imidazole, 1 mM phenylmethyl sulfonyl fluoride (PMSF)) plus EDTA-free protease inhibitor tablets (1 tablet per 50 mL, Roche, Cat. no. 05056489001), 0.5% Triton X-100, followed by homogenization with a dounce homogenizer and sonication (4 × 2000 J). The lysate was clarified by ultracentrifugation at 125,171 g (40,000 rpm in Beckman Ti45) for 30 min. The clarified lysate was then applied to a 10 mL bed volume Nickel nitrilotriacetic acid (Ni-NTA) agarose (Qiagen, Cat. no. 30230) column, washed with 10 column volumes of lysis buffer plus EDTA-free protease inhibitor tablets (1 tablet per 50 mL), 20 mM imidazole, 0.2% Triton X-100, followed by washing with 5 × CV of lysis buffer plus 20 mM imidazole. The protein was eluted with elution buffer (50 mM Tris, pH 8.0, 300 mM NaCl, 5% glycerol, 250 mM imidazole, 0.5 mM TECP, EDTA-free protease inhibitor tablets (1 tablet per 50 mL)). The protein was further purified by size exclusion chromatography with Superose 6 resin. The resulting purified KCK-VASP was eluted in storage buffer (25 mM HEPES pH 7.5, 200 mM NaCl, 5% glycerol, 1 mM EDTA, 5 mM DTT). Single-use aliquots were flash-frozen using liquid nitrogen and stored at −80 °C until the day of an experiment.

The mini-Lpd (his-Z-EGFP-LZ-Lpd(aa850-1250)) was transformed into *Escherichia coli* BL21 (NEB, Cat. no. C2527H) and grown at 30 °C to an OD of 0.8. Protein expression was performed using the following steps, which were adapted from a method previously described but with minor alterations[5]. The bacteria were then cooled to 12 °C and induced for 24 h with 1 mM IPTG. The rest of the protocol was performed at 4 °C. Cells were pelleted from a 2 L culture by centrifugation at 4785 g (5000 rpm in Beckman JLA-8.100) for 20 min. Pellets were resuspended in 100 mL of lysis buffer (50 mM sodium phosphate pH 8.0, 300 mM NaCl, 10 mM imidazole, 0.5 mM TCEP, 0.2% Triton X-100, 10% glycerol, 1 mM PMSF, and EDTA free protease inhibitor tablets (1 tablet per 50 mL) (Roche, Cat. no. 05056489001)) followed by sonication on ice for 4 × 2000 J with amplitude at 10 (Sonicator Qsonica LLC, Q700). The lysate was clarified by centrifugation at 48,384 g (20,000 rpm in Beckman JA25.50) for 30 min at 4 °C before being applied to a 10 mL bed volume Nickel nitrilotriacetic acid (Ni-NTA) agarose (Qiagen, Cat. no. 30230) column, and washed with 10 column volumes (CVs) of lysis buffer to which imidazole had been added to a final concentration of 20 mM. The column was then washed with 5 column volumes of lysis buffer containing 20 mM Imidazole but lacking Triton X-100 and protease inhibitor tablets. The protein was eluted with elution buffer (50 mM Tris-HCl, pH 7.5, 300 mM NaCl, 10% glycerol, 400 mM imidazole, 1 mM TECP, 1 mM PMSF, and EDTA-free protease inhibitor tablets (1 tablet per 50 mL). The protein was concentrated using Amicon Ultra-15, 30 K MWCO (Millipore, Cat. no. UFC903024) to 5 mL, and clarified by ultracentrifugation for 5 min at 68,000g (40,000 rpm with Beckman optimal MAX-E Ultracentrifuge and TLA100.3 rotor). The protein was further purified by size exclusion chromatography with Superose 6, and ion exchange chromatography with SP Sepharose Fast Flow (GE Healthcare, Cat. no. 17-0729-01), and stored as liquid nitrogen pellets at −80 °C.

Expression and purification of his-LAF-1 RGG, containing amino acids 1–168 of LAF-1 and a C-terminal 6xHis tag, was performed using the following steps, which were adapted from a method previously described, including starting plasmids[97]. In brief, RGG was overexpressed in *E. coli* BL21(DE3) cells. One-liter cultures were induced with 0.5 mM IPTG overnight at 18 °C and 220 rpm, and pellets of cells expressing his-LAF-1 RGG were harvested from the cultures via centrifugation when the OD 600 reached 0.8. The pellets were then lysed in a buffer containing 20 mM Tris, 500 mM NaCl, 20 mM imidazole, and one EDTA-free protease inhibitor tablet (Sigma-Aldrich) for 5 min on ice and then sonicated. The cell lysates were centrifuged at 134,000g for

40 min, after which his-LAF-1 RGG resided in the supernatant. The supernatant was then mixed with Ni-NTA resin (G-Biosciences) for 1 h, settled in a glass column, and washed with a buffer containing 20 mM Tris, 500 mM NaCl, 20 mM imidazole (pH 7.5). The bound proteins were eluted from the Ni-NTA resin with a buffer containing 20 mM Tris, 500 mM NaCl, 500 mM imidazole (pH 7.5). The purified proteins were then buffer exchanged into 20 mM Tris, 500 mM NaCl (pH 7.5) using Amicon Ultra centrifugal filters. Small aliquots of the protein were frozen in liquid nitrogen at a protein concentration of approximately 120 mM and stored at −80 °C. To promote solubility of the LAF-1 RGG protein, the entire purification process was performed at room temperature, except for the cell lysis process, which was done on ice.

**Protein labeling.** The VASP used in this study is a previously published "cysteine light" mutant that replaced the three endogenous cysteines with two serines and an alanine. A single cysteine was then introduced at the N-terminus of the protein to allow selective labeling with maleimide dyes. This mutant was found to function in an indistinguishable manner from the wild-type proteins[98]. Thus, VASP was labeled at the N-terminal cysteine using maleimide-conjugated dyes. VASP was buffer exchanged into 20 mM Tris (pH 7.4) 150 mM NaCl buffer to remove DTT from the storage buffer and then incubated with dye for 2 h at room temperature. Free dye was then removed by applying the labeling reaction to a spin column packed with Sephadex G-50 Fine DNA Grade (GE Healthcare GE17-0573-01) hydrated with TNT buffer (20 mM Tris pH 7.4, 150 mM NaCl, and 5 mM TCEP). The labeled protein was then centrifuged at 100,000g for 10 min at 4 °C to remove aggregates before being flash-frozen in single-use aliquots.

Monomeric actin was labeled using maleimide-conjugated dyes. Dyes were incubated with G-actin for 2 h at room temperature before being separated from the labeled protein by applying the labeling reaction to a spin column packed with Sephadex G-50 Fine DNA Grade (GE Healthcare GE17-0573-01) hydrated with a buffer (5 mM Tris-HCL (pH 8), 0.2 mM ATP, and 0.5 mM DTT, pH 8). The labeled protein was then centrifuged at 100,000g for 10 min at 4 °C to remove aggregates before being flash-frozen in single-use aliquots.

RGG was labeled using amine-reactive NHS Ester dyes. RGG was buffer exchanged into 25 mM HEPES (pH 7.4), 500 mM NaCl prior to labeling with dyes. After labeling, RGG was separated from unconjugated dye and buffer exchanged back to 20 mM Tris (pH 7.4) 500 mM NaCl buffer using 3 K Amicon columns (MilliporeSigma, Burlington, MA). Small aliquots of the protein were frozen in liquid nitrogen and stored at −80 °C.

**Protein droplet formation and actin polymerization.** Condensates composed of VASP, monomer mini-Lpd, or RGG were formed using the following steps, which were adapted from a method previously described with minor alterations[5,6]. The given concentrations of the respective proteins were mixed with 3% (w/v) PEG-8000 in 20 mM Tris, pH 7.4, 5 mM TCEP, and either 150 mM NaCl (for VASP condensates) or 50 mM NaCl (for monomer mini-Lpd and RGG condensates). PEG was added last to induce droplet formation after the protein was well-mixed in the solution. All protein concentrations refer to the monomeric concentrations. For actin polymerization assays within protein droplets, condensates were allowed to form for 10 min following PEG addition, after which Atto-594-labeled G-actin was mixed into the droplet solution. Following actin addition, the sample was incubated at room temperature for 15 min prior to imaging to allow for actin polymerization and assembly. For phalloidin-actin assays, unlabeled actin monomers were added to protein droplets and allowed to polymerize for 15 min. Phalloidin-iFluor594 (Abcam) was then added and allowed to stain filamentous actin for 15 min prior to imaging.

**Microscopy.** Samples were prepared for microscopy in 3.5 mm diameter wells formed using biopsy punches to create holes in 1.6 mm

thick silicone gaskets (Grace Biolabs) on Hellmanex III cleaned, no. 1.5 glass coverslips (VWR). Coverslips were passivated using poly-L-lysine conjugated PEG chains (PLL-PEG). After the addition of samples to these wells, an additional small coverslip was placed on top of the gasket to seal the imaging chamber and prevent evaporation during imaging. Fluorescence microscopy was done using an Olympus SpinSR10-Yokogawa SoRa spinning disk confocal microscope with a Hamamatsu Orca Flash 4.0V3 Scientific CMOS camera. FRAP was done using the Olympus FRAP unit 405 nm laser.

PLL-PEG was prepared using the following steps, which were adapted from a method previously described with minor alterations[5,99]. Amine-reactive mPEG succinimidyl valerate was conjugated to poly-L-lysine at a molar ratio of 1:5 PEG to PLL. The conjugation reaction takes place in a 50 mM sodium tetraborate solution, pH 8.5, and is allowed to react overnight at room temperature while continuously stirring. The final product is then buffer exchanged to PBS pH 7.4 using 7000 MWCO Zeba spin desalting columns (Thermo Fisher) and stored at 4 °C.

**Image analysis.** ImageJ was used to quantify the distribution of condensate characteristics. Specifically, condensates were selected using thresholding in the brightest channel and shape descriptors (i.e., diameter, aspect ratio, etc.), and protein fluorescent intensities were measured using the built-in analyze particles function. For aspect ratio analysis, condensates that had come into contact with other condensates were removed from the analysis to avoid any skewing of data from misrepresentation of single condensate deformation.

For actin ring thickness and power law fitting analysis, quantification of the ring thickness within the protein condensates was performed by measuring the phalloidin-actin intensity across a radial line bisecting the ring, taking the full-width at half-maximum intensity of the actin ring using a Gaussian fit of the intensity profile across one side of the actin ring, and then subtracting the diffraction limit to get an approximate ring thickness. To compare droplet diameters among aspherical droplets with AR > 1, we calculated an effective spherical diameter for the droplets based on their major and minor axis lengths: $D = \sqrt[3]{ab^2}$, where a refers to the major axis, and b refers to the minor axis lengths.

## Data analysis

**Filament length probability density function calculation.** The lengths of each filament are calculated by adding up the length of each segment at each time step. A filament length distribution is then created by sorting the lengths into 130 bins of equal width 0.05 μm. Since the filament length distribution changes with each time step, we then smooth the distribution over time by calculating a moving average of filament lengths over a period of ±5 time steps (±5 s) at each time step. Then we feed this into a kernel density estimation to form a Gaussian approximation of the probability density function of filament lengths at each timestep.

**Actin-covered surface fraction calculation.** As the actin filaments grow to form shells, discs, and rings, they will move to occupy space near the surface of the droplet. Since the droplet boundary is modeled as a dynamically deformable ellipsoid, the droplet shape may be different at each time point. As such, at each time point, we discretize the surface of the droplet to an icosphere that is then deformed according to the current state of the ellipsoid. The actin-covered surface fraction calculation then proceeds as in our previous study[39]. Actin filaments are discretized to the monomer level and, using a threshold distance of 0.1 μm from the surface of the deformed icosphere, each monomer is assigned to the nearest surface triangle. These assignments are then used to calculate the surface density via the fraction of occupied triangles at each time point.

**Filament alignment angle calculation.** Throughout the duration of our simulations, actin elongation within the confinement of a droplet drives the rearrangement of individual filaments, which leads to the emergence of actin network structures such as shells, discs, and rings. These structures differ in how filaments are organized with respect to one another. As such, we define an alignment angle as a metric that describes the extent to which the actin network structure has collapsed from a 3D spherical shell to a tight 2D ring, inclusive of intermediary shapes. At each time point, we first discretize the actin filaments to the monomer level and use the resulting filament positions to construct the gyration tensor for the filament network. We next extract the eigenvalues of the gyration tensor to compute the ellipsoidal spans, thus providing us with the axis lengths for an ellipsoidal shape approximation that best explains the distribution of filaments in the network. The alignment angle is thus defined as the angle between the major axis (span a) and minor axis (span c) of the ellipsoidal approximation. As such, the filament alignment angle varies from 0° (perfectly flat) to 45° (perfectly spherical).

**Statistics and reproducibility.** Due to the stochastic nature of the interactions in our simulations, 10 replicate simulations were performed per condition to yield a sufficiently large sample size for statistical analyses of the results. All Cytosim trajectories were initialized with a random distribution of filaments and crosslinker species. For power law fitting of simulation data, initial diameter values with fewer than 10 samples were excluded to limit the influence of sparsely sampled regions of the parameter space. This criterion was applied uniformly and affected 2 of 12 fits, from which only 2–3 data points per fit were excluded. All other fitting and analyses were performed using the full datasets. Experiments with condensates composed of VASP, monomer mini-Lpd, and RGG were repeated independently three times per condition. All data analysis for simulations was performed on Python 3.11.5.

### Reporting summary
Further information on research design is available in the Nature Portfolio Reporting Summary linked to this article.

## Data availability
The Cytosim input files and data generated in this study from experiments and simulations have been deposited in a Zenodo repository[100]. The data generated in this study is also available on GitHub (https://github.com/RangamaniLabUCSD/abp-deformable-droplet-Mansour-2025.git). Source data are provided with this paper.

## Code availability
The customized Cytosim source code used to generate trajectories and the Python scripts used to analyze the Cytosim trajectories have been deposited in a Zenodo repository[100]. The code is also available on GitHub (https://github.com/RangamaniLabUCSD/abp-deformable-droplet-Mansour-2025.git).

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

## Acknowledgements

This research was supported by the National Science Foundation through MODULUS Grants MCB 2327243 and MCB 2327244 to P.R and J.C.S., and by the National Institutes of Health through R35GM139531 to J.C.S. D.M. was supported by the Molecular Biophysics Training Grant, NIH Grant T32 GM139795. C.W. was supported through the Center for Dynamics and Control of Materials: an NSF MRSEC under Cooperative Agreement No. DMR-2308817. D.J. was supported by the Ronald E. McNair Scholars Program. C.T.L. was supported in part by a Kavli Institute for Brain and Mind Postdoctoral Fellowship.

## Author contributions

D.M., A.C., J.C.S., and P.R. designed the study. D.M., A.C., and C.T.L. designed the code and conducted simulations. C.W., D.J., and K.G. conducted the experiments. D.M., D.J., C.W., A.C., and K.G. analyzed the data. All authors wrote and edited the manuscript.

## Competing interests

P.R. is a consultant for Simula Research Laboratories in Oslo, Norway and receives income. The terms of this arrangement have been reviewed and approved by the University of California San Diego in accordance with its conflict-of-interest policies. The remaining authors declare no competing interests.
