## [Transparent Peer Review file · Nature Communications]

Mechanochemical feedback between confinement and actin crosslinking drives the shape dynamics of liquid-like droplets

Corresponding Author: Professor Padmini Rangamani

Version 0:

Reviewer comments:

Reviewer #1

(Remarks to the Author)

This work by Mansour et al., combines biochemistry, microscopy, and agent-based modeling to answer some interesting biophysical questions related to the organization of actin structures in the presence of actin-binding protein condensates. This work is incremental, building upon previous studies and observations made by the same group – as such, it builds on an established literature with significant implications for increasing our understanding of the general principles underlying cytoskeletal organization and the generation of complex actomyosin networks in biological and synthetic systems. While the paper is incremental, building on observations with rigid geometries, the introduction of deformable droplets in silico, along with the biochemical assays and imaging performed in this work do provide some significant advancement to our understanding of the biophysical mechanisms by which actin structures can assemble at different scales within cells. Furthermore, the authors mention modifications to the code of a commonly-used agent-based simulation engine, Cytosim, which is employed by many researchers working on cytoskeletal and cell biology questions. These modifications, which introduce dynamically multimerizing monomers based on a specified multimer formation and splitting rates, could be of benefit to the larger community of researchers working with Cytosim.

It is my recommendation that this manuscript be considered for publication, but only after some textual concerns are addressed. The only significant weakness I noticed with this work is that not enough detail is provided in the introduction to establish a biological context to the significance of the observations being made. The authors introduce that similar observations to theirs have been made for actin organization in GUVs (but the authors do not define for their reader what a GUV is) and fail to provide a context for actin-binding proteins forming biological condensates within cells until the end of the discussion... there, the authors reference work demonstrating that focal adhesion proteins can and do form condensates on membranes, which provides a clear context in which the organization of actin within droplets bears biological significance. Until this point, my reading of the paper was hindered as the interesting biophysical observations being made did not contain a clear tie-in to any real biology. I therefore strongly recommend that the initial reference to focal adhesion condensates be moved to the introduction.

Minor things to address:

1. "GUVs" are not defined in the introduction.

2. In the methods the authors mention their Cytosim code modifications and Python code will be made available upon publication. The code and scripts should be available already as reviewers may want or need to inspect it for validity. Further, this seems to contradict a statement from the supplemental methods which states: "The implementation of our dynamic multimerization model required edits to the Cytosim source code to simulate multimerization reactions and have previously been made available on a GitHub repository alongside a previous publication."

3. Italics needed for 'in vivo' and 'in vitro' in several places.

4. Confinement of actin within the condensate is an important component of these models and observations. In the in vitro condensate system actin is clearly confined. Is this true of any biological condensate? Also, according to the parameter table, actin confinement within the deformable space was 200 pN/um, is this based on some biological observation, or was there a parameter sweep performed to select this number? What would happen if this number was increased?

5. Figure 2D - A visual example of each PCM paradigm would be useful to understand what the morphological difference is between the four outcomes. Minimally, their occurrence in A could be highlighted by bordering the simulations with the same color scheme.
6. Besides focal adhesion proteins, do other actin-binders form condensates with a known purpose within cells?
7. "Actin rings appeared at the droplet periphery. Actin ring thickness within mini-Lpd condensates was quantified by Gaussian fitting of the fluorescence intensity profile across one side of each ring to measure the full-width at half-maximum, subtracting the diffraction limit to approximate the true thickness of the actin rings. The effective spherical condensate diameter for asymmetrical condensates was calculated by plotting ring thickness as a function of condensate diameter (Fig. 6D)." This is a method explanation in the results. Move to the methods section.
8. In Figure 6, why are there rings of mini-Lpd when neither VASP nor RGG also formed rings? This should be explained.
9. Figure 8C and 8D are too busy. These figures lack clarity and should probably be split into simpler panels.

Reviewer #2

(Remarks to the Author)

Mansour et al. follow up on their prior work that showed that actin binding proteins (like VASP, Lamellipodin and even synthetic chimeras) can form phase separated condensates that recruit G-actin and promote its assembly into F-actin, with or without intrinsic polymerization activity. Interestingly, depending on the experimental conditions and to some extent stochastically, the outcome of these experiments are very variable when filaments are allowed to grow beyond the droplet radius: initially spherical droplets may remain spherical with a shell of F-actin at the surface, deform into an oblate spheroid with a compact ring or somewhat less compact disk that aligns with its major axes or extend all the way into rod shape. Their prior work using simulations and in vitro experiments has already established that the length of actin filaments as well as the multivalency of the crosslinker VASP plays a role in determining the outcome via kinetic trapping. Here they add a layer of sophistication to their simulations by making droplets deformable with varying material properties. For the sphere to spheroid transition, this allows them to ask how droplet deformation affects actin networks within and inversely how actin networks with different properties affect droplet deformation kinetics. They find that the relationship between droplet size and actin bundle thickness generally follows a power law in vitro as well as in their simulations. At least in simulations, the droplet deformation follows an unexpected non-monotonous path in certain conditions during actin filament elongation. For as much as I can assess as a molecular biologist without much insight on simulations, the work seems methodologically correct. I however find that the interpretation and comparison of the theoretically derived, simulated and experimentally determined power laws is a bit superficial and thus adds little to what I might intuitively have expected. While the quality of the simulations seems impressive to me and the mechanics of this system are fascinating, the relevance for biology in general remains a bit vague.

Major Concerns:

1. One of the main claims in this work is that elongating actin filaments can cause non-monotonous deformation behavior while filaments grow. There seems to be no validation of that prediction, though. I assume that the precise actin filament length is difficult to assess in vitro, but if the time scale of filament growth on the order of minutes in the simulations is reproducible in vitro, an early aspect ratio "hump" with a magnitude of around 1.3 should be observable in time course experiments if it exists. In case it is not observed, is it because it is impossible to create condensates/crosslinkers with the predicted required properties or would it suggest that there is an unaccounted mechanism that allows bypassing the hump?
2. The other main claim of the manuscript is that filament number or bundle diameter required to deform a droplet as a function of droplet radius universally follows a power law. This is shown for simulations with rigid, dynamic and unspecific bundling as well as in vitro for the first two cases. The exponents match particularly well for the VASP/rigid crosslinking (considering a factor 2 between number and diameter), but much less so for the multivalent Lpd example. For the unspecific case, no in vitro data is shown. A theoretical derivation in the supplement yields a power law as well. It is, however, not entirely clear to me what the significance of power laws with rather variable exponents would be. I think this should be discussed in more detail. Is there anything we could learn from the coefficients or exponents or that it generally follows power laws? Could the conditions of the simulation be changed to match the power laws that were experimentally determined (e.g. in case of LPD, maybe for RGG or just variations of in vitro conditions) and would that allow us to learn something about the biochemistry of the condensates and crosslinkers? It feels like there might be some missed opportunities given the sophisticated simulation framework that the authors have built.
3. The biological relevance of this work is almost not at all covered in the text. The authors mention focal adhesions as an example where feedback between actin network and condensate deformation might play a role but remain extremely vague. It would help to elaborate more on why such feedback might matter in that case or other potential cases. Also, it might be good to mention in the introduction that condensates of VASP and other ABPs have been observed to recruit F-actin in cells (e.g. <https://doi.org/10.1038/s41586-022-05084-3> or <https://doi.org/10.1016/j.ejcb.2009.02.185> might be useful citations. I am not aware of in vivo examples of filaments bending along the condensate surface, but if the authors know any, that will help too.

Minor points:

1. In the legends of figures 2, 4 and 6 as well as table S2, the units of viscosity are given in pN s/ μm . Should it be pN s/ μm^2 as in table S1 or am I confused about the meaning of effective viscosity?
2. In the first paragraph on page 11: "However, when the bundles of actin grew longer than $\pi R \mu\text{m}$, the aspect ratio returned to growing monotonically" should it say $4R$ or $4 \pi R/3$ instead?
3. In figure 3 (or maybe a supplementary figure), would it be possible to convert the number of filaments to thickness in μm , such that the simulation data would be more comparable to the in vitro data?
4. In figure 6D, the power law fit does not look particularly convincing. It seems to me that it fits small droplets well, but since there are not so many that are bigger than $3 \mu\text{m}$, that area seems to not have contributed much to determining the fitted curve. Would there be a way of fitting the data that is not affected by the scarcity of the data points in a particular region of the plot? I think a better fit might give a power law with an exponent that is more comparable to the ones in the simulations.
5. In the same figure, are the actin densities (as measured by fluorescence intensity) in the thicker rings comparable to thinner rings in smaller droplets? If it is variable, might an integration of the fluorescent signal give a better approximation of filament number than thickness?
6. Fig. 7. Could similar power laws be determined for the in vitro data? It looks like there are less rings here. Do the droplets extend into rods without going through ring shaped intermediates? Is anything like that observed in the simulations with particular parameters and would that allow determining material properties of the condensates?

Version 1:

Reviewer comments:

Reviewer #1

(Remarks to the Author)

The authors undertook a significant rewrite of the introduction and discussion sections of the manuscript. More biological context was added to provide significance and improve interest from a wider audience. Overall, I believe the authors addressed all of my major concerns with the manuscript and took several suggestions on improving figure clarity and legibility. The resultant manuscript feels more polished and ready for acceptance.

(Remarks on code availability)

All the necessary code to run their modifications is there and appears to compile fine. Unfortunately, the way the authors branched from the main Cytosim branch, it is unclear where the major code modifications are. It might be useful for their commit comments to point to the specific files where modifications were made over "vanilla" Cytosim.

Reviewer #2

(Remarks to the Author)

I appreciate the authors' efforts to address all my previous comments and write a comprehensive response letter listing them in detail. While some of my major concerns were only partially fixed, the authors justified clearly why parts of my suggestions were hard to implement due to experimental limitations and adjusted the text accordingly. The added general biological context to the introduction and discussion is a great improvement in my opinion. I still sense a bit of a conceptual jump from the new part of the introduction to the motivation for the specific experiments performed in this manuscript and similarly I am missing a cohesive argument about how the results are relevant to the in vivo condensate examples that are now provided in the discussion. I think the manuscript can be published with minor edits, but the relevance to a biological audience could still be further improved by polishing those parts. Here are some specific things I noticed during rereading:

1. Lines 785-6: "Previous studies have shown that actin confinement within rigid spheres^{38,39} and inter-filament attraction are necessary conditions for bundling^{28,31}" To the best of my knowledge, actin filaments can be bundled without confinement. Did you mean to say "sufficient" or is this under specific circumstances?
2. Lines 823-4: I would mention more explicitly that one of the cited manuscripts observes actin organized along the surface of condensates, reminiscent of the simulations in this work.
3. Lines 826-49: I agree that FAs, their maturation from nascent dynamic into mature more layered structures and particularly how forces produced by the actin cytoskeleton contribute to that process are a potentially exciting case of mechanical functions of condensates. But how is this related to the work presented in this manuscript? I do not understand the sentence on lines 835-6 that suggests a connection.

(Remarks on code availability)

Mechanochemical feedback between confinement and actin crosslinking drives the shape dynamics of liquid-like droplets

Authors: Daniel Mansour, Dominique Jordan, Caleb Walker, Aravind Chandrasekaran, Christopher T. Lee, Kristin Graham, Jeanne Stachowiak, Padmini Rangamani

References associated with the response to the reviewers are located at the bottom of this document. Reference numbers within direct text from the revised manuscript correspond to the reference list in the manuscript document.

Reviewer #1:

1.1 This work by Mansour et al., combines biochemistry, microscopy, and agent-based modeling to answer some interesting biophysical questions related to the organization of actin structures in the presence of actin-binding protein condensates. This work is incremental, building upon previous studies and observations made by the same group – as such, it builds on an established literature with significant implications for increasing our understanding of the general principles underlying cytoskeletal organization and the generation of complex actomyosin networks in biological and synthetic systems. While the paper is incremental, building on observations with rigid geometries, the introduction of deformable droplets in silico, along with the biochemical assays and imaging performed in this work do provide some significant advancement to our understanding of the biophysical mechanisms by which actin structures can assemble at different scales within cells.

Author Response:

We thank the reviewer for their positive evaluation, which acknowledges that our work provides a significant advancement in our understanding of the biophysical mechanisms underlying actin structure assembly. We understand that the reviewer might perceive this work as possibly incremental because of our group's contribution to the field. However, this work is holistic because, through a series of systematically implemented computational models, we seek to identify shared principles of actin network organization in phase-separated droplets of actin-binding proteins. Indeed, we were quite pleased to note that the reviewer highlights the computational advances of the work, specifically the introduction of deformable droplets and the resulting significant advances that emerge from combining modeling and experiments. We have elaborated on the novelty of the work in the revision in addition to addressing all the reviewer's remarks.

1.2 Furthermore, the authors mention modifications to the code of a commonly-used agent-based simulation engine, Cytosim, which is employed by many researchers working on cytoskeletal and cell biology questions. These modifications, which introduce dynamically multimerizing monomers based on a specified multimer formation and splitting rates, could be of benefit to the larger community of researchers working with Cytosim.

Author Response:

We thank the reviewer for noting these significant technical improvements to Cytosim. We have made the code publicly available on GitHub (<https://github.com/RangamaniLabUCSD/abp-deformable-droplet-Mansour-2025.git>) so that this codebase is of use to the larger community.

1.3 It is my recommendation that this manuscript be considered for publication, but only after some textual concerns are addressed. The only significant weakness I noticed with this work is that not enough detail is provided in the introduction to establish a biological context to the significance of the observations being made. The authors introduce that similar observations to theirs have been made for actin organization in GUVs (but the authors do not define for their reader what a GUV is) and fail to provide a context for actin-binding proteins forming biological condensates within cells until the end of the discussion... there, the authors reference work demonstrating that focal adhesion proteins can and do form condensates on membranes, which provides a clear context in which the organization of actin within droplets bears biological significance. Until this point, my reading of the paper was hindered as the interesting biophysical observations being made did not contain a clear tie-in to any real biology. I therefore strongly recommend that the initial reference to focal adhesion condensates be moved to the introduction.

Author Response:

We have extensively revised the introduction to provide a biological context for our model system and findings. The changes to the text are as follows:

Biomolecular condensates form membraneless organelles that function as dynamic compartments for organizing various cellular functions¹⁻³. Recent discoveries have shown that several actin-binding proteins (ABPs) can form condensates, through liquid-liquid phase separation, both *in vitro* and *in vivo*³⁻¹². Over 160 ABPs have been identified, and each of them plays a distinct role in transforming actin monomers and unorganized filaments into specialized structures that drive complex biological processes^{13,14}. Despite their diversity, ABPs collectively regulate actin filaments through

behaviors that can be broadly classified as tuning filament assembly and disassembly rates, branching, severing, annealing, and bundling, which leads to the formation of a variety of actin network architectures. These structures provide the mechanical framework for filopodia, lamellipodia, stress fibers, focal adhesions, contractile rings, and the actin cortex^{4,13,15,16}. The reorganization of the actin cytoskeleton underlies a wide range of cellular processes that require cells to generate and transmit force, change shape, and respond to mechanical and chemical cues in their environment, including motility, signaling, and intracellular transport^{10,17–21}. Dysregulation of these processes is implicated in many pathologies, notably in cancer and metastasis²⁰. Importantly, ABP condensates have been observed in cells and are thought to provide a favorable environment for the recruitment of actin monomers, actin filaments, and other cytoskeletal proteins, and to promote nucleation of actin filaments^{12,16,22–27}. However, a comprehensive understanding of the role of actin condensates in actin network organization and remodeling is an active area of investigation.

Reconstituted systems have been employed to study interactions between actin and ABP condensates in a controlled environment to understand principles of actin organization and confinement²⁸. Similar actin ring formation has been observed in experimental studies using giant unilamellar vesicles (GUVs)^{29,30} and vesicles^{31–33}. Beyond the biochemical function of the constituent ABPs, the material and physical properties of condensates also regulate their function^{24,34,35}. The same weak interactions that drive condensate formation also allow for the development of the cohesive forces, which endow the liquid droplet with viscoelastic properties such as surface tension and viscosity^{24,35,36}. For actin filaments already within condensates, the surface tension drives confinement, which supports the accumulation of bending energy^{5,6,37}. Condensate deformation occurs once the filament bending energy exceeds the surface energy^{6,38}. Bending energy may accumulate from a combination of actin polymerization and increases in actin bundle rigidity mediated by ABP processes such as actin crosslinking. The resulting interplay between surface energy and bending energy plays a key role in regulating the resulting structures.

Reviewer Comment:

1. “GUVs” are not defined in the introduction.

Author Response:

We thank the reviewer for bringing this oversight to our attention. We have now expanded the definition of GUV in the text. The changes to the text are as follows:

Similar actin ring formation has been observed in experimental studies using giant unilamellar vesicles (GUVs)^{29,30} and vesicles^{31–33}.

Reviewer Comment:

2. In the methods the authors mention their Cytosim code modifications and Python code will be made available upon publication. The code and scripts should be available already as reviewers may want or need to inspect it for validity. Further, this seems to contradict a statement from the supplemental methods which states: “The implementation of our dynamic multimerization model required edits to the Cytosim source code to simulate multimerization reactions and have previously been made available on a GitHub repository alongside a previous publication.”

Author Response:

We have made the Cytosim code modifications and Python analysis scripts publicly available on a dedicated GitHub repository for this publication. We note that we had developed the dynamic multimerization model for a previous publication¹. In this work, we integrated the dynamic multimerization model¹ with the actin network model² and introduced capping kinetics. Finally, we adopted the deformable ellipsoidal boundary formulation from Dmitrieff et al. 2017³ to create a computational workflow that can capture all the necessary elements of the different cases considered here.

Action Taken:

We have added the Data Availability and Code Availability statements to the manuscript, which includes a link to the GitHub repository containing the Cytosim source code with the additional integrated functionality for this publication. The changes to the text are as follows:

Data Availability

The Cytosim input files and source data for experiments and simulations are available at <https://github.com/RangamaniLabUCSD/abp-deformable-droplet-Mansour-2025.git>.

Code Availability

The customized Cytosim source code used to generate trajectories and the Python scripts used to analyze the Cytosim trajectories are available on GitHub at <https://github.com/RangamaniLabUCSD/abp-deformable-droplet-Mansour-2025.git>.

The implementation of our dynamic multimerization model required edits to the Cytosim source code to simulate multimerization reactions, as detailed in a previous publication⁵.

Reviewer Comment:

3. Italics needed for 'in vivo' and 'in vitro' in several places.

Author Response:

We have now italicized each instance of *in vivo* and *in vitro* in the text. The changes to the text are as follows:

Recent discoveries have shown that several actin-binding proteins (ABPs) can form condensates, through liquid-liquid phase separation, both *in vitro* and *in vivo*³⁻¹².

In this work, we use a combination of agent-based modeling and *in vitro* biophysical experiments to systematically dissect how the biochemical and physical properties of liquid-like droplets of ABPs modulate actin filament organization and droplet shape.

Reviewer Comment:

- 4a. Confinement of actin within the condensate is an important component of these models and observations. In the *in vitro* condensate system actin is clearly confined. Is this true of any biological condensate?

Author Response:

The reviewer is right in that for the systems focused on in our study, actin is confined to the droplet. To our knowledge, while actin can be confined in many different biological condensates, we cannot say with certainty that any biological condensate will confine actin. As an example, recent studies of synapsin-actin interactions show that actin can both be confined within the droplet and also form networks that connect multiple droplets⁴. One advantage of the systems we are studying is that we can focus on actin networks that emerge due to confinement in droplets, allowing us to integrate modeling and experiments to draw insights. Ongoing work in our group is focused on understanding the conditions under which actin filaments may “escape” from a droplet made of actin-binding proteins.

Action Taken:

We have discussed other configurations of actin-droplet interactions in the discussion section of the revised manuscript. The changes to the text are as follows:

Biomolecular condensates are emerging as a key regulatory mechanism of cellular phenomena. The idea that containment of actin in a droplet can lead to feedback between bundling and condensate deformation can have implications in the organization of actin across a wide variety of biomolecular condensates. While we have shown such feedback in VASP condensates, these principles may be generalizable and

apply to other condensates containing actin. For example, synapsin, a protein responsible for regulating synaptic vesicle clustering in the presynapse of neurons, has been shown to form condensates that recruit and organize actin^{12,76}. Additionally, it has been recently shown in live cells that α -synuclein can regulate the material properties of synapsin droplets and alter their ability to cluster synaptic vesicles⁷⁷. While many condensates form spherical three-dimensional droplets, recent studies have shown that two-dimensional condensates can form on the surface of the plasma membrane where transmembrane functions necessitate coupling between the condensate and the membrane. For condensates that assemble on surfaces, such as in focal adhesions and tight junctions, wetting and prewetting interactions with the membrane emerge from interfacial energy minimization, which depends on the material properties and characteristics of the condensate and surface^{4,26,27,74,78–81}. These properties determine how the condensate spreads over the membrane, the droplet size, and the spatial patterning on the membrane^{75,82,83}. This feedback between bundling and condensate deformation shapes how actin networks assemble and remodel around focal adhesions in cells as well. Focal adhesion proteins such as talin and vinculin have been shown to form 2D condensate structures on the cellular membrane^{7,8,80}. The phase separation of focal adhesion and focal adhesion adapter proteins forms a dynamic and flexible assembly that clusters integrins and couples them to actin filaments, thus mediating the mechanical coupling between the actin cytoskeleton and the extracellular environment^{7,80}. Beyond confinement, in instances where the surface energy of the droplet is much smaller than the bending energy of the actin filaments, condensates can mediate actin crosslinking driven primarily by wetting filaments via capillary bridge interactions^{24,73,84,85}. Additionally, N-WASP forms condensates on the plasma membrane that function as focus points for Arp2/3 activation. N-WASP condensates form as a result of a prewetting transition whereby condensation occurs once N-WASP reaches a critical surface concentration⁷⁹. Thus, force generation at the membrane may result from feedback between droplet properties, actin bundling, and membrane properties^{86–88}.

Reviewer Comment:

4b. Also, according to the parameter table, actin confinement within the deformable space was 200 pN/ μm , is this based on some biological observation, or was there a parameter sweep performed to select this number? What would happen if this number was increased?

Author Response:

We originally chose 200 pN/ μm , as was done in our previous work^{1,2,5} and in Dmitrieff et al. 2017³. In response to this question, we conducted additional simulations for two different values of boundary repulsion stiffness (100 pN/ μm and 400 pN/ μm). Overall, for this range of values, the qualitative behavior of our observations did not change, but there were some quantitative differences. When we increased the actin confinement (400 pN/ μm), we observed greater droplet deformation in our simulations. We note that the role of the confinement term and the surface tension in Cytosim is different. The boundary repulsion stiffness is the parameter in the simulation that determines the strength of the inward-facing repulsive force applied to the actin filament to enforce the droplet boundary. An equal and opposite outward-facing force is then incorporated as a point force acting on the boundary. The point forces are balanced by the forces associated with surface tension, and subjected to volume incompressibility, in order to determine ellipsoid deformation. As such, for a fixed surface tension, a larger boundary repulsion stiffness increases the magnitude of the point forces, and thus deformation. For this reason, we keep the boundary repulsion stiffness constant throughout all of our simulations, as we are interested in the material properties of the condensate.

Action Taken:

We have included a supplementary figure that compares what happens when we increase or decrease the actin boundary repulsion stiffness in deformable VASP condensates under conditions that favor the formation of **A)** Discs and **B)** Rings. The changes to the text are as follows:

Point forces are calculated from the force exerted to keep actin filaments within the boundary, as modulated by the boundary repulsion stiffness (**Supplementary Fig. 1**).

This specifies the spring stiffness that acts on the discretized points of each actin filament and crosslinking molecule if the point lies outside the specified boundary. The force on each point is dependent on the distance that it lies beyond the confines of the boundary. Additionally, for actin filaments, these forces are used to calculate the point forces acting on the boundary that drive droplet deformation⁵¹.

Supplementary Figure 1: Magnitude of condensate aspect ratio varies with filament boundary repulsion stiffness. A) Time series showing the mean (solid line) and standard deviation (shaded area) of droplet aspect ratio for each actin boundary repulsion condition for simulations of VASP that have disc-forming kinetics ($k_{\text{bind}} = 0.1 \text{ s}^{-1}$, $k_{\text{unbind}} = 1.0 \text{ s}^{-1}$). **B)** Time series showing the mean (solid line) and standard deviation (shaded area) of droplet aspect ratio for each actin boundary repulsion condition for simulations of VASP that have ring-forming kinetics ($k_{\text{bind}} = 10.0 \text{ s}^{-1}$, $k_{\text{unbind}} = 1.0 \text{ s}^{-1}$). Representative final snapshots of each condition are included above each plot.

Reviewer Comment:

- Figure 2D - A visual example of each PCM paradigm would be useful to understand what the morphological difference is between the four outcomes. Minimally, their occurrence in A could be highlighted by bordering the simulations with the same color scheme.

Author Response:

We thank the reviewer for their suggestions on how to improve Figure 2D. We have made modifications to include representative snapshots for each of the four clusters we identified from our principal component analysis. Additionally, we have included a new Supplementary Table 3, which details the clusters that correspond to each VASP kinetic condition. The changes to the text are as follows:

D) K-means clustering was performed on the last five snapshots from each replicate (data shown in **B** and **C**), which revealed four cluster categories corresponding to actin structures that are discs, rings, loose shells, or tight shells. The data for each snapshot is projected onto a scatterplot of the first two principal components (PC1 and PC2), and colored by clusters. Representative snapshots of each cluster are included above the plot. Please see Supplementary Table 3 for the corresponding cluster identities for each combination of kinetic parameters.

The actin network shapes formed by each combination of kinetic parameters are given in **Supplementary Table 3**.

Supplementary Table 3: Table of actin network shapes for each (k_{bind} , k_{unbind}) condition

k_{bind} (1/s)	10.0	Tight Shell	Loose Shell	10% Loose Shell ----- 90% Ring	Ring	Ring
	1.0	Tight Shell	90% Loose Shell ----- 10% Ring	Ring	Ring	Disc
	0.1	80% Tight Shell ----- 20% Loose Shell	Loose Shell	Ring	Disc	Disc
	0.01	90% Tight Shell ----- 10% Loose Shell	20% Ring ----- 80% Disc	Disc	Disc	Disc
	0.001	Disc	Disc	Disc	Disc	Disc
		0.001	0.01	0.1	1.0	10.0
		k_{unbind} (1/s)				

Reviewer Comment:

6. Besides focal adhesion proteins, do other actin-binders form condensates with a known purpose within cells?

Author Response:

Yes, many actin-binding proteins form condensates that serve a known function within cells. For example, as mentioned in the response to this reviewer's comment 4a, synapsin forms condensates that help regulate the synaptic vesicle cluster in the presynapse of neurons. Additionally, N-WASP forms condensates on the plasma membrane that function as focus points for Arp2/3 activation.

Action Taken:

We have expanded the introduction and discussion to provide greater biological context and add additional references to other condensates that bind actin.

Reviewer Comment:

7. "Actin rings appeared at the droplet periphery. Actin ring thickness within mini-Lpd condensates was quantified by Gaussian fitting of the fluorescence intensity profile across one side of each ring to measure the full-width at half-maximum, subtracting the diffraction limit to approximate the true thickness of the actin rings. The effective spherical condensate diameter for asymmetrical condensates was calculated by plotting ring thickness as a function of condensate diameter (Fig. 6D)." This is a method explanation in the results. Move to the methods section.

Author Response:

We have removed this passage from the results section and ensured its placement in the image analysis section of the supplementary methods. The changes to the text are as follows:

Actin rings appeared at the droplet periphery. The thickness of the actin ring and the effective diameter of the encapsulating mini-Lpd condensate were quantified as described in the **Supplementary Methods**. We then plotted ring thickness as a function of effective condensate diameter and extracted a power law scaling relationship (Fig. 6D).

Reviewer Comment:

8. In Figure 6, why are there rings of mini-Lpd when neither VASP nor RGG also formed rings? This should be explained.

Author Response:

We believe that the reviewer is referring to Figure 6C. We would like to thank the reviewer for pointing out this lack of clarification. The “ring” like shape that is seen in the mini-Lpd condition, we define as a toroidal morphology due to the evacuation of the condensate phase from the center of the actin ring. We have a recent publication that explains that these toroidal shapes likely occur due to the balance between nucleation and elongation of actin filaments⁶. Toroidal morphologies appear to arise when there is less nucleation of new filaments, such that the actin filaments are trapped in a ring state, and the condensate deforms to accommodate the thickening actin ring. Due to actin being in a ring state, the filaments are unable to exert asymmetric pressure on the condensate boundary. As such, the sphere collapses into a disc and then a toroid rather than undergoing deformation into a rod-like morphology. This toroidal morphology is very infrequent in the conditions shown (<1% of condensates), and toroidal structures were not included in the ring thickness analysis.

Action Taken:

To avoid confusion for the readers, we have updated the images in Figure 6C with ones that are more representative of our analysis and lack a toroidal shape, as shown below:

Reviewer Comment:

9. Figure 8C and 8D are too busy. These figures lack clarity and should probably be split into simpler panels.

Author Response:

We thank the reviewer for making this suggestion and have made changes to simplify the panels and improve clarity.

Action Taken:

We have removed extraneous elements, improved the clarity of the plotted points, and split the panels such that the scatter plots and line plots have their own separate panels. The changes to the text are as follows:

Figure 8: Feedback between droplet deformation and actin filament alignment. **A)** Alignment angle for rigid ellipsoidal droplets with fixed but varied aspect ratios. The error bars represent the standard deviation. Inset images are final representative snapshots of each condition. **B)** Phase diagram plotting the final alignment angle and final aspect ratio across all replicates for simulations without crosslinkers. Each σ_{surface} condition is indicated by color. The inset graph shows the time series profile showing the mean (solid line) and standard deviation (shaded area) of the filament alignment angle (blue) and the droplet aspect ratio (red) for the condition indicated by a red circle ($\sigma_{\text{surface}} = 4 \text{ pN}/\mu\text{m}$). **C)** Phase diagram plotting the final alignment angle and final aspect ratio across all replicates for simulations with dynamically multimerizing crosslinkers. Each k_{form} condition is indicated by color, and each k_{split} condition is indicated by marker shape. **D)** Time series profile showing the mean (solid line) and standard deviation (shaded area) of the filament alignment angle (blue) and the droplet aspect ratio (red) for the condition indicated by the red circle in panel **C** ($k_{\text{form}} = 10.0 \text{ s}^{-1}$, $k_{\text{split}} = 10.0 \text{ s}^{-1}$). **E)** Phase diagram plotting the final alignment angle and final aspect ratio across all replicates for simulations with VASP crosslinkers. Each k_{bind} condition is indicated by color, and each k_{unbind} condition is indicated by marker shape. **F)** Time series profile showing the mean (solid line) and standard deviation (shaded area) of the filament alignment angle (blue) and the droplet aspect ratio (red) for the condition indicated by the red circle in panel **E** ($k_{\text{bind}} = 10.0 \text{ s}^{-1}$, $k_{\text{unbind}} = 10.0 \text{ s}^{-1}$).

Reviewer #2:

Mansour et al. follow up on their prior work that showed that actin binding proteins (like VASP, Lamellipodin and even synthetic chimeras) can form phase separated condensates that recruit G-actin and promote its assembly into F-actin, with or without intrinsic polymerization activity. Interestingly, depending on the experimental conditions and to some extent stochastically, the outcome of these experiments are very variable when filaments are allowed to grow beyond the droplet radius: initially spherical droplets may remain spherical with a shell of F-actin at the surface, deform into an oblate spheroid with a compact ring or somewhat less compact disk that aligns with its major axes or extend all the way into rod shape. Their prior work using simulations and in vitro experiments has already established that the length of actin filaments as well as the multivalency of the crosslinker VASP plays a role in determining the outcome via kinetic trapping. Here they add a layer of sophistication to their simulations by making droplets deformable with varying material properties. For the sphere to spheroid transition, this allows them to ask how droplet deformation affects actin networks within and inversely how actin networks with different properties affect droplet deformation kinetics. They find that the relationship between droplet size and actin bundle thickness generally follows a power law in vitro as well as in their simulations. At least in simulations, the droplet deformation follows an unexpected non-monotonous path in certain conditions during actin filament elongation. For as much as I can assess as a molecular biologist without much insight on simulations, the work seems methodologically correct. I however find that the interpretation and comparison of the theoretically derived, simulated and experimentally determined power laws is a bit superficial and thus adds little to what I might intuitively have expected. While the quality of the simulations seems impressive to me and the mechanics of this system are fascinating, the relevance for biology in general remains a bit vague.

Author Response:

We thank the reviewer for their positive evaluation of our work and for acknowledging the sophistication of our simulations. In the revised version, and as noted in the response to Reviewer 1, we have extensively edited the introduction and the discussion to emphasize the biological significance of our work.

Major Concerns:

Reviewer Comment:

1. One of the main claims in this work is that elongating actin filaments can cause non-monotonous deformation behavior while filaments grow. There seems to be no validation of that prediction, though. I assume that the precise actin filament length is difficult to assess in vitro, but if the time scale of filament growth on the order of minutes in the simulations is reproducible in vitro, an early aspect ratio “hump” with a magnitude of around 1.3 should be observable in time course experiments if it exists. In case it is not observed, is it because it is impossible to create condensates/crosslinkers with the predicted required properties or would it suggest that there is an unaccounted mechanism that allows bypassing the hump?

Author Response:

Unfortunately, due to technical limitations associated with the image acquisitions in the experiments, we have not yet been able to visualize droplet deformation dynamics that are non-monotonic. To ensure that the light exposure does not cause the condensates to form gels and completely alter their material properties, we restrict our imaging to intervals of 10-15 s. Since extended exposure causes the droplets to become stiffer, our attempts to acquire a time-lapse of droplet deformation have only worked for small droplets, where the non-monotonicity of deformation would be difficult to resolve. However, in the simulations, we do not have such constraints. An additional element at play here is that we cannot measure or control for the number of actin filaments internalized by the droplets in experiments, but there are likely hundreds of filaments within each droplet. In contrast, in the simulations, we do not simulate hundreds of filaments to ensure that the system does not blow up out of combinatorial complexity. While it is possible that the hump in the data is a computational artifact, similar dynamic snapping behaviors are well known to occur in viscoelastic systems undergoing mechanical deformations⁷⁻¹¹.

Action Taken:

We have noted these points as a limitation of our work in the discussion. The changes to the text are as follows:

While current imaging modalities were not able to capture the snapping behavior in experiments, the common principle driving droplet deformation dynamics across all cases we studied is that the droplet deforms when the energy exerted by the filament bundle overcomes the interfacial energy of the droplet.

Reviewer Comment:

2a. The other main claim of the manuscript is that filament number or bundle diameter required to deform a droplet as a function of droplet radius universally follows a power law. This is shown for simulations with rigid, dynamic and unspecific bundling as well as in vitro for the first two cases. The exponents match particularly well for the VASP/rigid crosslinking (considering a factor 2 between number and diameter), but much less so for the multivalent Lpd example. For the unspecific case, no in vitro data is shown.

Author Response:

We agree with the reviewer that there was a larger discrepancy for the power law fits with the mini-Lpd conditions. Therefore, we conducted additional experiments to obtain more data for larger mini-Lpd droplets. We then refit the data to the equation $y = ax^b$ and obtained a better fit ($R^2 = 0.8251$) and an exponent ($b = 1.9910$) that is more comparable to the simulation results as well as the results from previous experiments.

In response to this reviewer's comment, we also attempted to quantify ring thicknesses for the nonspecific case. However, a combination of the RGG condensate properties and passive physical filament bundling resulted in actin rings that were too diffuse to provide highly quantitative data across a broad range of droplet diameters, especially for larger condensate diameters, as would be required to map the power law dependency for this case.

Action Taken:

We updated Figure 6D with the additional data and new power law fit. We have noted our inability to resolve a clear trend in RGG condensates as a minor limitation of our study. The changes to the text are as follows:

D) Power law scaling relation between the measured actin ring thickness and the diameter of monomer mini-Lpd condensates ($R^2 = 0.8251$, $n = 228$). The inset images correspond to the denoted points on the graph and depict examples of the difference in measured actin ring thickness. Scale bars, 1 μm .

We note that the RGG droplets were diffuse at larger condensate diameters. As ring thicknesses across a range of droplet diameters are necessary to derive a power law trend, we were unable to quantify the relationship between droplet diameter and ring thickness in RGG droplets.

Reviewer Comment:

2b. A theoretical derivation in the supplement yields a power law as well. It is, however, not entirely clear to me what the significance of power laws with rather variable exponents would be. I think this should be discussed in more detail. Is there anything we could learn from the coefficients or exponents or that it generally follows power laws?

Author Response:

The power laws throughout the manuscript show that as the droplet size increases, the thickness of the actin ring required to deform the droplet increases nonlinearly according to a power law $y = ax^b$, where a is the prefactor and b is the exponent. This general principle holds true across all cases tested in experiments and simulations, and is also consistent with experiments in other systems¹². From the simulations, we note that increasing the surface tension of the droplet increases the prefactor (Fig. 3B, Fig. 3D, Fig. 6B, and Fig. 7B), which represents the number of filaments required to deform

a droplet with an initial diameter of 1.0 μm , according to the mathematical fit of the simulation data. As such, across all of our simulations, we observe that higher surface tension droplets require a greater number of filaments, and therefore greater force, to deform to the same extent as lower surface tension droplets. The nonlinear relation between the number of filaments and the initial diameter of the droplets is described by the exponent and affected by the properties of the crosslinkers. However, interpreting trends in the values of the exponent alone is trickier. If we were to fix the value of the prefactor at 1.0 ($a = 1.0$), then we would expect to see the exponent increase as we increase the surface tension. However, fixing the prefactor is an unrealistic scenario because more filaments are required to deform stiffer droplets. As such, it is a combination of the prefactor and the exponent that describes the data. Beyond these observations, we do not have the experimental or computational resolution to draw insights into the specific values of the exponents at this time.

Action Taken:

We have included a more detailed discussion of the significance of the power laws obtained from simulations in the discussion section of the revised manuscript. The changes to the text are as follows:

In both cases, the number of filaments needed to deform droplets scales as a power law with respect to droplet radius, indicating that larger droplets need more forces from actin filaments to deform. This general principle is true across all the cases tested in experiments and in simulations, and is also consistent with experiments in other systems²⁹. The prefactor of the power law depends on the interfacial properties of the droplet. Increasing the surface tension increases the prefactor, indicating that, for droplets with high surface tension, a higher number of filaments, and therefore a greater force, is needed to deform a droplet of the same diameter.

Reviewer Comment:

- 2c.** Could the conditions of the simulation be changed to match the power laws that were experimentally determined (e.g. in case of LPD, maybe for RGG or just variations of in vitro conditions) and would that allow us to learn something about the biochemistry of the condensates and crosslinkers? It feels like there might be some missed opportunities given the sophisticated simulation framework that the authors have built.

Author Response:

As noted in response to comment 2b, the entire power law formulation indicates a nonlinear scaling between the thickness of the actin ring and droplet diameter. The prefactor represents the number of filaments required to deform a droplet with an initial

diameter of 1.0 μm , while the exponents describe the nonlinear relation and are affected by the properties of the crosslinkers. As such, the terms of the power law are representative of the material and biochemical properties of the droplet and constituent crosslinkers. Since force generation in our model depends on the filaments, we can, in principle, generate predictions that link surface tension in simulations to experiments. However, we must acknowledge that the complexity of the physical reality of the condensate environment limits our ability to generate precise values. Our simulations are based upon a minimal physics model of actin crosslinking and organization in deformable condensates, where deformation is implemented with a simple boundary deformation model restricted to ellipsoidal geometries. Additionally, our model does not explicitly represent molecular crowding or include hydrodynamic interactions. In a real biomolecular condensate, variations in local interfacial conditions may result in varied surface tension along the interface and a variety of more complex geometries. While we appreciate the idea of matching the exponents between the simulations and experiments, we note that this is a rather challenging exercise in parameter variation due to its combinatorial complexity. We agree that there is an opportunity to explore this further in the future.

Action Taken:

We have noted these points as a limitation of our work in the discussion. The changes to the text are as follows:

While our findings have broad implications for the interactions between the cytoskeleton and protein droplets, we acknowledge that we have constructed a simple model that reproduces the minimal physics of actin organization within condensates. To improve upon our current simulations, future models require the incorporation of hydrodynamic interactions between the actin filaments and the fluid droplet^{89–92}.

Reviewer Comment:

- 3a. The biological relevance of this work is almost not at all covered in the text. The authors mention focal adhesions as an example where feedback between actin network and condensate deformation might play a role but remain extremely vague. It would help to elaborate more on why such feedback might matter in that case or other potential cases.

Author Response:

As noted in our response to Reviewer 1, we have extensively revised the introduction and discussion of our manuscript to address the biological relevance of our work.

The changes to the introduction are as follows:

Biomolecular condensates form membraneless organelles that function as dynamic compartments for organizing various cellular functions¹⁻³. Recent discoveries have shown that several actin-binding proteins (ABPs) can form condensates, through liquid-liquid phase separation, both *in vitro* and *in vivo*³⁻¹². Over 160 ABPs have been identified, and each of them plays a distinct role in transforming actin monomers and unorganized filaments into specialized structures that drive complex biological processes^{13,14}. Despite their diversity, ABPs collectively regulate actin filaments through behaviors that can be broadly classified as tuning filament assembly and disassembly rates, branching, severing, annealing, and bundling, which leads to the formation of a variety of actin network architectures. These structures provide the mechanical framework for filopodia, lamellipodia, stress fibers, focal adhesions, contractile rings, and the actin cortex^{4,13,15,16}. The reorganization of the actin cytoskeleton underlies a wide range of cellular processes that require cells to generate and transmit force, change shape, and respond to mechanical and chemical cues in their environment, including motility, signaling, and intracellular transport^{10,17-21}. Dysregulation of these processes is implicated in many pathologies, notably in cancer and metastasis²⁰. Importantly, ABP condensates have been observed in cells and are thought to provide a favorable environment for the recruitment of actin monomers, actin filaments, and other cytoskeletal proteins, and to promote nucleation of actin filaments^{12,16,22-27}. However, a comprehensive understanding of the role of actin condensates in actin network organization and remodeling is an active area of investigation.

Reconstituted systems have been employed to study interactions between actin and ABP condensates in a controlled environment to understand principles of actin organization and confinement²⁸. Similar actin ring formation has been observed in experimental studies using giant unilamellar vesicles (GUVs)^{29,30} and vesicles³¹⁻³³. Beyond the biochemical function of the constituent ABPs, the material and physical properties of condensates also regulate their function^{24,34,35}. The same weak interactions that drive condensate formation also allow for the development of the cohesive forces, which endow the liquid droplet with viscoelastic properties such as surface tension and viscosity^{24,35,36}. For actin filaments already within condensates, the surface tension drives confinement, which supports the accumulation of bending energy^{5,6,37}. Condensate deformation occurs once the filament bending energy exceeds the surface energy^{6,38}. Bending energy may accumulate from a combination of actin polymerization and increases in actin bundle rigidity mediated by ABP processes such as actin crosslinking. The resulting interplay between surface energy and bending energy plays a key role in regulating the resulting structures.

The changes to the discussion are as follows:

Biomolecular condensates are emerging as a key regulatory mechanism of cellular phenomena. The idea that containment of actin in a droplet can lead to feedback between bundling and condensate deformation can have implications in the organization of actin across a wide variety of biomolecular condensates. While we have shown such feedback in VASP condensates, these principles may be generalizable and apply to other condensates containing actin. For example, synapsin, a protein responsible for regulating synaptic vesicle clustering in the presynapse of neurons, has been shown to form condensates that recruit and organize actin^{12,76}. Additionally, it has been recently shown in live cells that α -synuclein can regulate the material properties of synapsin droplets and alter their ability to cluster synaptic vesicles⁷⁷. While many condensates form spherical three-dimensional droplets, recent studies have shown that two-dimensional condensates can form on the surface of the plasma membrane where transmembrane functions necessitate coupling between the condensate and the membrane. For condensates that assemble on surfaces, such as in focal adhesions and tight junctions, wetting and prewetting interactions with the membrane emerge from interfacial energy minimization, which depends on the material properties and characteristics of the condensate and surface^{4,26,27,74,78–81}. These properties determine how the condensate spreads over the membrane, the droplet size, and the spatial patterning on the membrane^{75,82,83}. This feedback between bundling and condensate deformation shapes how actin networks assemble and remodel around focal adhesions in cells as well. Focal adhesion proteins such as talin and vinculin have been shown to form 2D condensate structures on the cellular membrane^{7,8,80}. The phase separation of focal adhesion and focal adhesion adapter proteins forms a dynamic and flexible assembly that clusters integrins and couples them to actin filaments, thus mediating the mechanical coupling between the actin cytoskeleton and the extracellular environment^{7,80}. Beyond confinement, in instances where the surface energy of the droplet is much smaller than the bending energy of the actin filaments, condensates can mediate actin crosslinking driven primarily by wetting filaments via capillary bridge interactions^{24,73,84,85}. Additionally, N-WASP forms condensates on the plasma membrane that function as focus points for Arp2/3 activation. N-WASP condensates form as a result of a prewetting transition whereby condensation occurs once N-WASP reaches a critical surface concentration⁷⁹. Thus, force generation at the membrane may result from feedback between droplet properties, actin bundling, and membrane properties^{86–88}.

Reviewer Comment:

3b. Also, it might be good to mention in the introduction that condensates of VASP and other ABPs have been observed to recruit F-actin in cells (e.g. <https://doi.org/10.1038/s41586-022-05084-3> or <https://doi.org/10.1016/j.ejcb.2009.02.185> might be useful citations. I am not aware of in vivo examples of filaments bending along the condensate surface, but if the authors know any, that will help too.

Author Response:

We agree with the reviewer that it is important to mention F-actin recruitment into condensates of ABPs in cells. We thank the reviewer for recommending additional important references to include in our manuscript. We have now included these references as reference 22 and reference 23.

Action Taken:

As noted in our response to this reviewer's major concern 3a, we have made extensive edits to the introduction and discussion. Additionally, we have included an explicit mention in the introduction that ABP condensates recruit actin filaments in cells. The changes to the text are as follows:

Importantly, ABP condensates have been observed in cells and are thought to provide a favorable environment for the recruitment of actin monomers, actin filaments, and other cytoskeletal proteins, and to promote nucleation of actin filaments^{12,16,22–27}.

Minor points:

Reviewer Comment:

1. In the legends of figures 2, 4 and 6 as well as table S2, the units of viscosity are given in pN s/ μm . Should it be pN s/ μm^2 as in table S1 or am I confused about the meaning of effective viscosity?

Author Response:

Essentially, there are two different viscosity parameters in our simulations. The viscosity referenced in Supplementary Table 1, with units pN s/ μm^2 , describes the resistance to displacement felt by diffusing species within the droplet medium and sets the timescale of diffusion within our simulations. The effective viscosity ($\mu_{\text{effective}}$) that is referenced in Figure 2, Figure 4, Figure 6, and Supplementary Table 2 is a distinct parameter that describes the resistance to changes in the displacement of the three axes of the deformable ellipsoid and sets the timescale of droplet surface deformation. The units for

$\mu_{\text{effective}}$ are worked out from the formula for the time evolution of deformation, described in Dmitrieff et al. 2017³ as,

$$\dot{a}_k = \frac{1}{\mu} \left(\phi_p^k + \phi_\sigma^k + \Sigma \phi_f^k \right) \quad (1)$$

where, for each direction k , \dot{a}_k is the rate of deformation (units $\mu\text{m/s}$) and ϕ^k are the effective forces (units pN). This relation yields the units of pN s/ μm for $\mu_{\text{effective}}$.

Action Taken:

We have added text to clarify the distinction between $\mu_{\text{effective}}$ and the droplet viscosity. The changes to the text are as follows:

Time evolution casts the force balance along the three axes of the ellipsoid and calculates the speed at which deformation occurs as modulated by the effective viscosity, $\mu_{\text{effective}}$, which acts as a damping parameter to slow the deformation of the ellipse axes over time without affecting the final droplet shape⁵¹. **Note that $\mu_{\text{effective}}$ (units pN s/ μm) sets the timescale of droplet deformation and is a distinct parameter from the droplet viscosity (units pN s/ μm^2), which we hold constant at 0.5 pN s/ μm^2 throughout all simulations.** A full formulation of this model can be found in Dmitrieff et al. 2017⁵¹.

Reviewer Comment:

2. In the first paragraph on page 11: “However, when the bundles of actin grew longer than $\pi R \mu\text{m}$, the aspect ratio returned to growing monotonically” should it say $4R$ or $4 \pi R/3$ instead?

Author Response:

We thank the reviewer for pointing out this error. The monotonic increase in aspect ratio resumed at around 400 s, which corresponds with an actin filament length of about $4\pi R/3 \mu\text{m}$. The text has been updated to reflect the correct length of actin filaments. The changes to the text are as follows:

However, when the bundles of actin grew longer than $4\pi R/3 \mu\text{m}$, the aspect ratio returned to growing monotonically (**Fig. 3A, Supplementary Fig. 4**).

Reviewer Comment:

3. In figure 3 (or maybe a supplementary figure), would it be possible to convert the number of filaments to thickness in μm , such that the simulation data would be more comparable to the in vitro data?

Author Response:

Unfortunately, we were unable to convert the number of filaments to a thickness in the simulations. Unlike previous 2D simulations¹³, increasing the number of filaments does not necessarily scale the thickness of the rings. In 3D, filaments can be arranged and packed in numerous ways to form a variety of actin structures that are not necessarily uniform in thickness or even ring-like in shape. As such, determinations of ring thickness in our simulations would not give a good quantitative match with experiments. However, common to both of these simulations is that the forces generated by actin scale with the number of filaments. For these reasons, we chose the number of actin filaments in simulations as the most comparable to the ring thicknesses measured in our experiments.

Reviewer Comment:

4. In figure 6D, the power law fit does not look particularly convincing. It seems to me that it fits small droplets well, but since there are not so many that are bigger than 3 μm , that area seems to not have contributed much to determining the fitted curve. Would there be a way of fitting the data that is not affected by the scarcity of the data points in a particular region of the plot? I think a better fit might give a power law with an exponent that is more comparable to the ones in the simulations.

Author Response:

In response to this question, we conducted additional experiments and refit the data to obtain a better power law fit that is more comparable to results from simulations and previous experiments. Additional details are included in the response to this reviewer's major concern 2a. The revised figure is included below:

D) Power law scaling relation between the measured actin ring thickness and the diameter of monomer mini-Lpd condensates ($R^2 = 0.8251$, $n = 228$). The inset images correspond to the denoted points on the graph and depict examples of the difference in measured actin ring thickness. Scale bars, 1 μm .

Reviewer Comment:

5. In the same figure, are the actin densities (as measured by fluorescence intensity) in the thicker rings comparable to thinner rings in smaller droplets? If it is variable, might an integration of the fluorescent signal give a better approximation of filament number than thickness?

Author Response:

The reviewer is correct that an integration under the Gaussian fit would likely give a better approximation of the actin density within rings. However, in our experiments with phalloidin-stained actin filaments, phalloidin is added to the top of the imaging well after the condensate-actin mix is added in order to allow for actin assembly and settling to the coverslip. This leads to variations in the intensity of the phalloidin channel across the well as the phalloidin mixes and diffuses through the sample. Due to such variation, we are not confident that the integration of the Gaussian intensity profile would be completely accurate as a comparison tool; therefore, we have used the thickness of the actin rings instead.

Reviewer Comment:

6. Fig. 7. Could similar power laws be determined for the in vitro data? It looks like there are less rings here. Do the droplets extend into rods without going through ring shaped intermediates? Is anything like that observed in the simulations with particular parameters and would that allow determining material properties of the condensates?

Author Response:

In response to this reviewer's comment, we attempted to quantify ring thicknesses for the nonspecific case shown in Figure 7. However, the passive physical filament bundling that occurred using the nonspecific crosslinkers resulted in actin rings that were too diffuse at larger condensate diameters to provide highly quantitative data across a broad range of droplet diameters. As such, we were unable to obtain the data needed to identify a power law trend from RGG condensates. Accordingly, we have added a qualification to the text, which is cited below for ease of reference.

We note that the RGG droplets were diffuse at larger condensate diameters. As ring thicknesses across a range of droplet diameters are necessary to derive a power law trend, we were unable to quantify the relationship between droplet diameter and ring thickness in RGG droplets.

References

1. Walker, C. *et al.* Liquid-like condensates that bind actin promote assembly and bundling of actin filaments. *Dev. Cell* (2025) doi:10.1016/j.devcel.2025.01.012.
2. Chandrasekaran, A., Graham, K., Stachowiak, J. & Rangamani, P. Kinetic trapping organizes actin filaments within liquid-like protein droplets. *Nature Communications* **15**, 3139 (2024).
3. Dmitrieff, S., Alsina, A., Mathur, A. & Nédélec, F. J. Balance of microtubule stiffness and cortical tension determines the size of blood cells with marginal band across species. *Proc. Natl. Acad. Sci. U. S. A.* **114**, 4418–4423 (2017).
4. Chhabra, A. *et al.* Condensates of synaptic vesicles and synapsin-1 mediate actin sequestering and polymerization. *EMBO J.* **44**, 5112–5148 (2025).
5. Graham, K. *et al.* Liquid-like condensates mediate competition between actin branching and bundling. *Proc. Natl. Acad. Sci. U. S. A.* **121**, e22309152121 (2023).
6. Walker, C. *et al.* A balance between nucleating and elongating actin filaments controls deformation of protein condensates. *bioRxiv* 2025.06.18.660423 (2025) doi:10.1101/2025.06.18.660423.
7. Hassinger, J. E., Oster, G., Drubin, D. G. & Rangamani, P. Design principles for robust vesiculation in clathrin-mediated endocytosis. *Proc. Natl. Acad. Sci. U. S. A.* **114**, E1118–E1127 (2017).
8. Gomez, M., Moulton, D. E. & Vella, D. Dynamics of viscoelastic snap-through. *J. Mech. Phys. Solids* **124**, 781–813 (2019).
9. Gomez, M., Moulton, D. E. & Vella, D. Critical slowing down in purely elastic ‘snap-through’ instabilities. *Nat. Phys.* **13**, 142–145 (2017).
10. Mahapatra, A. & Rangamani, P. Formation of protein-mediated bilayer tubes is governed by a snapthrough transition. *Soft Matter* **19**, 4345–4359 (2023).
11. Alimohamadi, H., Ovryn, B. & Rangamani, P. Modeling membrane nanotube morphology: the role of heterogeneity in composition and material properties. *Sci. Rep.* **10**, 2527 (2020).
12. Limozin, L., Bärmann, M. & Sackmann, E. On the organization of self-assembled actin networks in giant vesicles. *Eur. Phys. J. E Soft Matter* **10**, 319–330 (2003).
13. Graham, K. *et al.* Liquid-like VASP condensates drive actin polymerization and dynamic bundling. *Nature Physics* **19**, 574–585 (2023).

Mechanochemical feedback between confinement and actin crosslinking drives the shape dynamics of liquid-like droplets

Authors: Daniel Mansour, Dominique Jordan, Caleb Walker, Aravind Chandrasekaran, Christopher T. Lee, Kristin Graham, Jeanne Stachowiak, Padmini Rangamani

Reference numbers within direct text from the revised manuscript correspond to the reference list in the manuscript document.

Reviewer #1:

1.1 The authors undertook a significant rewrite of the introduction and discussion sections of the manuscript. More biological context was added to provide significance and improve interest from a wider audience. Overall, I believe the authors addressed all of my major concerns with the manuscript and took several suggestions on improving figure clarity and legibility. The resultant manuscript feels more polished and ready for acceptance.

Author Response:

We thank the reviewer for their positive evaluation and for their acknowledgement of our extensive edits. We agree that the quality of the manuscript has been greatly improved by providing greater biological context and improving figure clarity.

1.2 All the necessary code to run their modifications is there and appears to compile fine. Unfortunately, the way the authors branched from the main Cytosim branch, it is unclear where the major code modifications are. It might be useful for their commit comments to point to the specific files where modifications were made over "vanilla" Cytosim.

Author Response:

We have imported the commit history to the GitHub repository, which includes the modifications made by the authors as well as the preexisting development that produced the underlying Cytosim version. In addition, we have included a README.md file in the Source_Code directory that points to the specific files where the major modifications build over base Cytosim. The contents of this file are provided below:

Modified Cytosim Source Code

Origin

The base Cytosim version can be found at <https://gitlab.com/f-nedelec/cytosim>. The modifications introduced by the authors to the dynamic_ellipse fork of Cytosim are documented in the commit history from 2024 onward and build upon the extensive preexisting development that produced the underlying Cytosim version.

Capping Protein

Capping protein was implemented implicitly as a special type of growing fiber introduced in this work. The following files are the major source code modifications:

- Cytosim/src/sim/fibers/capping_fiber.cc
- Cytosim/src/sim/fibers/capping_fiber.h
- Cytosim/src/sim/fibers/capping_fiber_prop.cc
- Cytosim/src/sim/fibers/capping_fiber_prop.h

Dynamic Ellipsoidal Deformation

The implementation of dynamic ellipsoidal deformation was built as part of the dynamic_ellipse fork of the base Cytosim version and presented in Dmitrieff et al. 2017. The following files are the major modifications that introduced the simple deformable boundary to Cytosim:

- Cytosim/src/sim/spaces/space_dynamic_ellipse.cc
- Cytosim/src/sim/spaces/space_dynamic_ellipse.h
- Cytosim/src/sim/spaces/space_dynamic_prop.cc
- Cytosim/src/sim/spaces/space_dynamic_prop.h

Dynamic Multimerization

Dynamically multimerizing crosslinkers was implemented as an extension of the dynamic dimers introduced in Walker et al. 2025. The following files are the major source code modifications:

- Cytosim/src/sim/hands/dimerizer.cc
- Cytosim/src/sim/hands/dimerizer.h
- Cytosim/src/sim/hands/dimerizer_prop.cc
- Cytosim/src/sim/hands/dimerizer_prop.h

Reviewer #2:

2.1 I appreciate the authors' efforts to address all my previous comments and write a comprehensive response letter listing them in detail. While some of my major concerns were only partially fixed, the authors justified clearly why parts of my suggestions were hard to implement due to experimental limitations and adjusted the text accordingly. The added general biological context to the introduction and discussion is a great improvement in my opinion.

Author Response:

We thank the reviewer for their positive evaluation of the extensive modifications made to the text. We agree that the quality of the manuscript has been greatly improved by addressing the previous remarks. We appreciate the reviewer for understanding our experimental limitations.

2.2 I still sense a bit of a conceptual jump from the new part of the introduction to the motivation for the specific experiments performed in this manuscript and similarly I am missing a cohesive argument about how the results are relevant to the *in vivo* condensate examples that are now provided in the discussion. I think the manuscript can be published with minor edits, but the relevance to a biological audience could still be further improved by polishing those parts. Here are some specific things I noticed during rereading:

Author Response:

We thank the reviewer for their feedback and have made edits to the introduction and discussion in response. In essence, we wanted to communicate that *in vivo* condensate systems are very complex, and our approach has been to build a minimal model that captures the key elements of actin organization within phase-separated droplets using a combination of *in vitro* and *in silico* systems.

The changes to the text of the introduction are as follows:

Due to the high complexity of *in vivo* condensate assemblies, reconstituted systems have been employed to study interactions between actin and ABP condensates in a controlled environment to understand the underlying biophysical principles of actin organization and confinement²⁸.

By combining *in vitro* and *in silico* approaches, we have been able to create a complementary framework where experiments provide physical constraints and simulations systematically explore the underlying mechanisms and generate new experimentally testable hypotheses. We have previously established how the spatial

confinement of ABPs through phase separation affects actin filament assembly and bundling^{5,6,39,40}.

However, two questions remain: What is the role of mechanical feedback from deformable boundary confinement in tuning actin organization within the droplet? How do the biochemical properties of the crosslinker influence droplet deformation? To address these questions, we built a minimal model that captures the key elements of actin organization in phase-separated droplets using a combination of computational and experimental approaches.

The changes to the text of the discussion are as follows:

The feedback between bundling and condensate deformation presented in this work may have implications for how surface-associated condensates respond to mechanical load, where shape changes and interfacial geometry are important for force transmission. In focal adhesions, proteins such as talin and vinculin form dynamic and flexible two-dimensional condensate assemblies on the cellular membrane that cluster integrins and couple them to actin filaments, thus mediating the mechanical coupling between the actin cytoskeleton and the extracellular environment^{7,8,82}. These assemblies undergo maturation under tension that is driven by actin filaments, which is accompanied by the reorganization of the actin architecture and changes in condensate shape, suggesting a mechanically analogous coupling between actin-generated forces and condensate deformation. Beyond confinement, in instances where the surface energy of the droplet is much smaller than the bending energy of the actin filaments, condensates can mediate actin crosslinking driven primarily by wetting filaments via capillary bridge interactions^{24,75,86,87}. Additionally, N-WASP forms condensates on the plasma membrane that function as focus points for Arp2/3 activation. N-WASP condensates form as a result of a prewetting transition whereby condensation occurs once N-WASP reaches a critical surface concentration⁸¹. Together, these examples illustrate how feedback between condensate material properties, actin organization, and membrane properties may contribute to force generation and cytoskeletal remodeling at the membrane⁸⁸⁻⁹⁰.

Reviewer Comment:

1. Lines 785-6: "Previous studies have shown that actin confinement within rigid spheres^{38,39} and inter-filament attraction are necessary conditions for bundling^{28,31}" To the best of my knowledge, actin filaments can be bundled without confinement. Did you mean to say "sufficient" or is this under specific circumstances?

Author Response:

We thank the reviewer for this suggestion. We agree that "sufficient" is a better choice of words in the context of the surrounding discussion. The changes to the text are as follows:

Previous studies have shown that actin confinement within rigid spheres^{39,40} and inter-filament attraction are **sufficient** conditions for bundling^{28,31}.

Reviewer Comment:

2. Lines 823-4: I would mention more explicitly that one of the cited manuscripts observes actin organized along the surface of condensates, reminiscent of the simulations in this work.

Author Response:

We thank the reviewer for this suggestion. We agree that more explicitly highlighting the similarities between these works strengthens the applicability of our findings to other condensate systems. The changes to the text are as follows:

For example, synapsin, a **presynaptic** protein **that** regulates synaptic vesicle clustering, has been shown to form condensates that recruit and organize actin **along the condensate surface in a manner similar to the condensates described in this work**^{12,78}.

Reviewer Comment:

3. Lines 826-49: I agree that FAs, their maturation from nascent dynamic into mature more layered structures and particularly how forces produced by the actin cytoskeleton contribute to that process are a potentially exciting case of mechanical functions of condensates. But how is this related to the work presented in this manuscript? I do not understand the sentence on lines 835-6 that suggests a connection.

Author Response:

We thank the reviewer for their feedback. We have revised the text to clarify the conceptual connection between focal adhesions and the feedback between actin bundling and condensate deformation identified here, specifically that focal adhesions respond to mechanical load. This comparison is intended to highlight focal adhesion maturation under tension and accompanying actin reorganization as mechanically analogous to the coupling between actin-generated forces, cytoskeletal remodeling, and condensate deformation. The changes to the text are as follows:

While many condensates form spherical three-dimensional droplets, recent studies have shown that two-dimensional condensates can form on the surface of the plasma membrane where transmembrane functions necessitate coupling between the condensate and the membrane. For condensates that assemble on surfaces, such as in focal adhesions and tight junctions, wetting and prewetting interactions with the membrane emerge from interfacial energy minimization, which depends on the material properties and characteristics of the condensate and surface^{4,26,27,76,80–83}. These properties determine how the condensate spreads over the membrane, the droplet size, and the spatial patterning on the membrane^{77,84,85}. The feedback between bundling and condensate deformation presented in this work may have implications for how surface-associated condensates respond to mechanical load, where shape changes and interfacial geometry are important for force transmission. In focal adhesions, proteins such as talin and vinculin form dynamic and flexible two-dimensional condensate assemblies on the cellular membrane that cluster integrins and couple them to actin filaments, thus mediating the mechanical coupling between the actin cytoskeleton and the extracellular environment^{7,8,82}. These assemblies undergo maturation under tension that is driven by actin filaments, which is accompanied by the reorganization of the actin architecture and changes in condensate shape, suggesting a mechanically analogous coupling between actin-generated forces and condensate deformation. Beyond confinement, in instances where the surface energy of the droplet is much smaller than the bending energy of the actin filaments, condensates can mediate actin crosslinking driven primarily by wetting filaments via capillary bridge interactions^{24,75,86,87}. Additionally, N-WASP forms condensates on the plasma membrane that function as focus points for Arp2/3 activation. N-WASP condensates form as a result of a prewetting transition whereby condensation occurs once N-WASP reaches a critical surface concentration⁸¹. Together, these examples illustrate how feedback between condensate material properties, actin organization, and membrane properties may contribute to force generation and cytoskeletal remodeling at the membrane^{88–90}.